# Clinical outcomes and cost-effectiveness of COVID-19 vaccination in South Africa

Krishna P. Reddy [1,2,3✉], Kieran P. Fitzmaurice[1], Justine A. Scott[1], Guy Harling [4,5,6,7,8], Richard J. Lessells [9], Christopher Panella[1], Fatma M. Shebl[1,3], Kenneth A. Freedberg[1,3,10,11,12] & Mark J. Siedner [1,3,4,11]

Low- and middle-income countries are implementing COVID-19 vaccination strategies in light of varying vaccine efficacies and costs, supply shortages, and resource constraints. Here, we use a microsimulation model to evaluate clinical outcomes and cost-effectiveness of a COVID-19 vaccination program in South Africa. We varied vaccination coverage, pace, acceptance, effectiveness, and cost as well as epidemic dynamics. Providing vaccines to at least 40% of the population and prioritizing vaccine rollout prevented >9 million infections and >73,000 deaths and reduced costs due to fewer hospitalizations. Model results were most sensitive to assumptions about epidemic growth and prevalence of prior immunity to SARS-CoV-2, though the vaccination program still provided high value and decreased both deaths and health care costs across a wide range of assumptions. Vaccination program implementation factors, including prompt procurement, distribution, and rollout, are likely more influential than characteristics of the vaccine itself in maximizing public health benefits and economic efficiency.

[1] Medical Practice Evaluation Center, Massachusetts General Hospital, Boston, MA, USA. [2] Division of Pulmonary and Critical Care Medicine, Massachusetts General Hospital, Boston, MA, USA. [3] Harvard Medical School, Boston, MA, USA. [4] Africa Health Research Institute, KwaZulu-Natal, South Africa. [5] MRC/Wits Rural Public Health and Health Transitions Research Unit (Agincourt), University of the Witwatersrand, Johannesburg, South Africa. [6] School of Nursing and Public Health, College of Health Sciences, University of KwaZulu-Natal, KwaZulu-Natal, South Africa. [7] Institute for Global Health, University College London, London, UK. [8] Department of Epidemiology and Harvard Center for Population and Development Studies, Harvard T.H. Chan School of Public Health, Boston, MA, USA. [9] KwaZulu-Natal Research Innovation and Sequencing (KRISP), College of Health Sciences, University of KwaZulu-Natal, Durban, South Africa. [10] Division of General Internal Medicine, Massachusetts General Hospital, Boston, MA, USA. [11] Division of Infectious Diseases, Massachusetts General Hospital, Boston, MA, USA. [12] Department of Health Policy and Management, Harvard T.H. Chan School of Public Health, Boston, MA, USA. ✉email: kpreddy@mgh.harvard.edu

The development and licensure of coronavirus disease 2019 (COVID-19) vaccines offer a critically important opportunity to curtail the global COVID-19 pandemic[1–4]. Even before the efficacy and safety of the leading vaccine candidates were established, many high-income countries (HICs) pre-emptively procured stocks of doses in excess of population need[5]. By contrast, most low- and middle-income countries (LMICs) do not have access to sufficient quantities of vaccine due to cost, limitations in available doses, and logistical challenges of production, distribution, and storage[6]. Meanwhile, the Africa Centres for Disease Control and Prevention have announced a goal of vaccinating 60% of Africans by the end of 2022[7].

There has been much discussion about reported efficacies and costs of different vaccines. However, factors specific to implementation, including vaccine supply, vaccination pace, and acceptance among communities, are increasingly recognized to be crucial to the effectiveness of a vaccination program in promoting epidemic control in HICs—in some cases, even more so than vaccine efficacy[8–11]. How these program implementation factors will affect the clinical and health economic consequences of COVID-19 in LMICs has not been well-defined. This is a particularly urgent question given the emergence of severe acute respiratory syndrome coronavirus 2 (SARS-CoV-2) variants, such as B.1.351 in South Africa, which appear to partially reduce efficacy of some vaccines[4,12–15].

In this work, we use a microsimulation model to estimate the clinical and economic outcomes of COVID-19 vaccination programs in South Africa, examining different implementation strategies that policymakers could directly influence. We simulate COVID-19-specific outcomes over 360 days, including daily and cumulative infections (detected and undetected), deaths, years-of-life lost (YLL) attributable to COVID-19 mortality, resource utilization (hospital and intensive care unit [ICU] bed use), and health care costs from the all-payer (public and private) health sector perspective. We examine different strategies of vaccination program implementation under multiple scenarios of vaccine effectiveness and epidemic growth, thereby projecting which factors have the greatest impact on clinical and economic outcomes and cost-effectiveness. Our goal was to inform vaccination program priorities in South Africa and other LMICs.

## Results

**Clinical and economic benefits of vaccination strategies.** To understand the trade-offs inherent to policy decisions regarding the total vaccine supply to purchase and the speed with which to administer vaccinations, we compared the clinical and economic outcomes of different strategies of population coverage (vaccine supply) and vaccination pace. We determined the incremental cost-effectiveness ratio (ICER) of each strategy as the difference in health care costs (2020 USD) divided by the difference in years-of-life saved (YLS) compared with other strategies of supply and pace. We considered multiple scenarios of epidemic growth, including a scenario in which the effective reproduction number ($R_e$) varies over time to produce two waves of SARS-CoV-2 infections.

In both the $R_e = 1.4$ scenario and the two-wave epidemic scenario, the absence of a vaccination program resulted in the most infections (~19–21 million) and deaths (70,400–89,300), and highest costs (~$1.69–1.77 billion) over the 360-day simulation period (Table 1). Vaccinating 40% of the population decreased deaths (82–85% reduction) and resulted in the lowest total health care costs (33–45% reduction) in both scenarios. Increasing the vaccinated population to 67%, the government's target for 2021, decreased deaths and raised costs in both scenarios. Increasing the vaccine supply to 80%, while simultaneously increasing vaccine

acceptance to 80%, reduced deaths and raised costs even further in both scenarios. In the $R_e = 1.4$ scenario, the 67% supply strategy was less efficient (had a higher ICER) than the 80% supply strategy and the latter had an ICER of $4270/YLS compared with the 40% supply strategy. In the two-wave epidemic scenario, the 67% and 80% supply strategies had ICERs of $1,990/YLS and $2,600/YLS, respectively. A vaccine supply of 20%, while less efficient than higher vaccine supply levels, still reduced deaths by 72–76% and reduced costs by 15–32% compared with no vaccination. The highest vaccination pace, 300,000 vaccinations daily, resulted in the most favorable clinical outcomes and lowest costs compared with lower paces in both the $R_e = 1.4$ and the two-wave epidemic scenarios (Table 1).

Supplementary Table 1 details the differences between a reference vaccination program (supply 67%, pace 150,000 vaccinations/day) and no vaccination program in age-stratified cumulative infections and deaths, hospital and ICU bed use, and health care costs. The reference vaccination program reduced hospital bed days by 67% and ICU bed days by 54% compared with no vaccination program.

When varying both vaccine supply and vaccination pace across different scenarios of epidemic growth ($R_e$), a faster vaccination pace decreased both COVID-19 deaths and total health care costs, whereas the impact of a higher vaccine supply on deaths and costs varied (Table 1 and Supplementary Table 2). In all four $R_e$ scenarios, a vaccination strategy with supply 40% and pace 300,000/day resulted in fewer deaths and lower costs than a strategy with higher supply (67%) and slower pace (150,000/day). At a vaccination pace of 300,000/day, increasing the vaccine supply from 40% to 67% was cost-saving in the two-wave epidemic scenario, whereas it resulted in ICERs of $520/YLS when $R_e = 1.4$, $1160/YLS when $R_e = 1.8$, and $85290/YLS when $R_e = 1.1$.

**Sensitivity analysis: vaccine characteristics and alternative scenarios.** To understand the influence of extrinsic factors (i.e., those outside the direct control of vaccination program decision-makers, such as vaccine effectiveness and costs and epidemic growth), we performed sensitivity analyses in which we varied each of these factors. In each alternative scenario, we projected clinical and economic outcomes and determined the ICER of a reference vaccination program (67% vaccine supply, 150,000 vaccinations/day, similar to stated goals in South Africa) compared with no vaccination program[16–18].

In one-way sensitivity analysis, the reference vaccination program remained cost-saving compared with a scenario without vaccines across different values of effectiveness against infection, effectiveness against mild/moderate disease, effectiveness against severe/critical disease, and vaccine acceptance (Table 2). When increasing the cost per person vaccinated up to $25, the vaccination program remained cost-saving. At cost per person vaccinated between $26 and $75, the vaccination program increased health care costs compared with a scenario without vaccines, but the ICERs increased only to $1500/YLS (Table 2).

The reference vaccination program had an ICER < $100/YLS or was cost-saving compared with a scenario without vaccines across different values of prior immunity (up to 40%), initial prevalence of active COVID-19, reduction in transmission rate among vaccinated but infected individuals, and costs of hospital and ICU care (Table 2 and Supplementary Table 3). When there was 50% prior immunity, the vaccination program still reduced deaths but it increased costs, with an ICER of $22,460/YLS compared with a scenario without vaccines. Notably, when excluding costs of hospital care and ICU care, and only considering costs of the vaccination program, the program increased costs, but its ICER compared with no vaccination program was only $450/YLS

**Table 1 Clinical and economic outcomes of different COVID-19 vaccination program strategies of vaccine supply and vaccination pace under different scenarios of epidemic growth in South Africa.**

| Scenario and vaccination strategy | Cumulative SARS-CoV-2 infections | Cumulative COVID-19 deaths | Years-of-life lost | Health care costs, USD | ICER, USD per year-of-life saved[a] |
|---|---|---|---|---|---|
| **Vaccine supply** | | | | | |
| **$R_e = 1.4$** | | | | | |
| Vaccine supply 40% | 11,784,700 | 16,000 | 275,800 | 1,177,742,900 | -- |
| Vaccine supply 67% | 10,585,100 | 14,700 | 259,600 | 1,338,803,500 | Dominated |
| Vaccine supply 80%[b] | 10,410,000 | 12,000 | 217,900 | 1,425,272,800 | 4270 |
| Vaccine supply 20% | 15,489,500 | 21,800 | 397,300 | 1,508,890,800 | Dominated |
| No vaccination | 21,012,100 | 89,300 | 1,558,700 | 1,766,856,200 | Dominated |
| **Two-wave epidemic[c]** | | | | | |
| Vaccine supply 40% | 7,758,800 | 10,600 | 175,100 | 927,247,000 | -- |
| Vaccine supply 67% | 5,594,000 | 7,800 | 133,700 | 1,009,741,300 | 1990 |
| Vaccine supply 80%[b] | 5,940,500 | 6,900 | 119,100 | 1,047,885,500 | 2600 |
| Vaccine supply 20% | 12,765,900 | 19,900 | 371,500 | 1,148,772,700 | Dominated |
| No vaccination | 19,290,400 | 70,400 | 1,206,200 | 1,691,805,000 | Dominated |
| **Vaccination pace** | | | | | |
| **$R_e = 1.4$** | | | | | |
| Pace 300,000 vaccinations per day | 5,659,400 | 7,200 | 120,300 | 1,016,586,100 | -- |
| Pace 200,000 vaccinations per day | 8,191,900 | 9,600 | 151,300 | 1,123,694,300 | Dominated |
| Pace 150,000 vaccinations per day | 10,585,100 | 14,700 | 259,600 | 1,338,803,500 | Dominated |
| No vaccination | 21,012,100 | 89,300 | 1,558,700 | 1,766,856,200 | Dominated |
| **Two-wave epidemic[c]** | | | | | |
| Pace 300,000 vaccinations per day | 2,697,100 | 3,200 | 49,300 | 780,133,600 | -- |
| Pace 200,000 vaccinations per day | 4,148,500 | 5,900 | 90,300 | 881,291,000 | Dominated |
| Pace 150,000 vaccinations per day | 5,594,000 | 7,800 | 133,700 | 1,009,741,300 | Dominated |
| No vaccination | 19,290,400 | 70,400 | 1,206,200 | 1,691,805,000 | Dominated |
| **Vaccine supply and vaccination pace** | | | | | |
| **$R_e = 1.4$** | | | | | |
| Vaccine supply 40%, pace 300,000 vaccinations per day | 9,866,800 | 13,000 | 211,300 | 969,576,100 | -- |
| Vaccine supply 67%, pace 300,000 vaccinations per day | 5,659,400 | 7,200 | 120,300 | 1,016,586,100 | 520 |
| Vaccine supply 40%, pace 150,000 vaccinations per day | 11,784,700 | 16,000 | 275,800 | 1,177,742,900 | Dominated |
| Vaccine supply 67%, pace 150,000 vaccinations per day | 10,585,100 | 14,700 | 259,600 | 1,338,803,500 | Dominated |
| No vaccination | 21,012,100 | 89,300 | 1,558,700 | 1,766,856,200 | Dominated |
| **Two-wave epidemic[c]** | | | | | |
| Vaccine supply 67%, pace 300,000 vaccinations per day | 2,697,100 | 3,200 | 49,300 | 780,133,600 | -- |
| Vaccine supply 40%, pace 300,000 vaccinations per day | 6,223,600 | 7,200 | 126,900 | 780,274,900 | Dominated |
| Vaccine supply 40%, pace 150,000 vaccinations per day | 7,758,800 | 10,600 | 175,100 | 927,247,000 | Dominated |
| Vaccine supply 67%, pace 150,000 vaccinations per day | 5,594,000 | 7,800 | 133,700 | 1,009,741,300 | Dominated |
| No vaccination | 19,290,400 | 70,400 | 1,206,200 | 1,691,805,000 | Dominated |

Dominated: the strategy results in a higher ICER than that of a more clinically effective strategy, or the strategy results in less clinical benefit (more years-of-life lost) and higher health care costs than an alternative strategy. ICER incremental cost-effectiveness ratio, $R_e$ effective reproduction number, USD United States dollars.

aWithin each $R_e$ scenario, vaccination strategies are ordered from the lowest to highest cost per convention of cost-effectiveness analysis. ICERs are calculated compared to the next least expensive, non-dominated strategy. Displayed life-years and costs are rounded to the nearest hundred, whereas ICERs are calculated based on non-rounded life-years and costs, and then rounded to the nearest ten.

bWhen modeling a vaccination program that seeks to vaccinate 80% of the population, uptake among those eligible was increased to 80% of the population, to avoid a scenario in which supply exceeds uptake. If uptake is not increased beyond 67%, then the strategy of vaccinating 67% of the population provides the most clinical benefit and results in an ICER of $9,960/YLS compared with vaccinating 40% of the population when $R_e$ is 1.4 and $1,990/YLS in an epidemic scenario with periodic surges.

cIn the analysis of an epidemic with periodic surges, the basic reproduction number ($R_0$) alternates between low and high values over time, and the $R_e$ changes day-to-day as the epidemic and vaccination program progress, and there are fewer susceptible individuals. For most of the simulation horizon, $R_0$ is 1.6 (equivalent to an initial $R_e$ of 1.1). However, during days 90–150 and 240–300 of the simulation, $R_0$ is increased to 2.6. This results in two epidemic waves with peak $R_e$ of ~1.4–1.5.

**Table 2 One-way sensitivity analyses of different COVID-19 vaccine characteristic and epidemic growth scenarios in South Africa.**

| Parameter/value | SARS-CoV-2 infections averted, compared with no vaccination | COVID-19 deaths averted, compared with no vaccination | Years-of-life saved, compared with no vaccination | Change in health care costs, compared with no vaccination, USD | ICER, compared with no vaccination, USD per YLS[a] |
|---|---|---|---|---|---|
| Vaccine effectiveness in preventing SARS-CoV-2 infection, % | | | | | |
| 20 | 5,466,500 | 71,600 | 1,254,900 | −166,032,500 | Cost-saving |
| 40 (Base case) | 10,427,000 | 74,600 | 1,299,100 | −428,052,700 | Cost-saving |
| 50[b] | 12,758,000 | 77,500 | 1,349,700 | −554,501,500 | Cost-saving |
| 75[b] | 16,067,300 | 82,000 | 1,429,400 | −750,946,700 | Cost-saving |
| Vaccine effectiveness in preventing mild/moderate COVID-19, %[c] | | | | | |
| 29 | 8,310,500 | 74,000 | 1,298,900 | −377,101,700 | Cost-saving |
| 51 (Base case) | 10,427,000 | 74,600 | 1,299,100 | −428,052,700 | Cost-saving |
| 67 | 10,625,200 | 76,200 | 1,332,300 | −410,883,200 | Cost-saving |
| 79 | 10,722,500 | 75,300 | 1,316,800 | −399,131,600 | Cost-saving |
| Vaccine effectiveness in preventing severe or critical COVID-19 requiring hospitalization, %[d] | | | | | |
| 40 | 10,659,300 | 65,800 | 1,180,100 | −80,901,300 | Cost-saving |
| 86 (Base case) | 10,427,000 | 74,600 | 1,299,100 | −428,052,700 | Cost-saving |
| 98 | 10,690,200 | 77,500 | 1,341,700 | −545,358,200 | Cost-saving |
| Vaccine acceptance among those eligible, % | | | | | |
| 50 | 10,026,700 | 71,100 | 1,251,600 | −272,592,000 | Cost-saving |
| 67 (Base case) | 10,427,000 | 74,600 | 1,299,100 | −428,052,700 | Cost-saving |
| 90 | 10,562,300 | 79,200 | 1,360,000 | −526,334,700 | Cost-saving |
| Vaccination cost per person, USD | | | | | |
| 9 | 10,427,000 | 74,600 | 1,299,100 | −656,846,300 | Cost-saving |
| 14.81 (Base case) | 10,427,000 | 74,600 | 1,299,100 | −428,052,700 | Cost-saving |
| 25 | 10,427,000 | 74,600 | 1,299,100 | −26,778,000 | Cost-saving |
| 26 | 10,427,000 | 74,600 | 1,299,100 | 12,601,200 | 10 |
| 35 | 10,427,000 | 74,600 | 1,299,100 | 367,014,600 | 280 |
| 45 | 10,427,000 | 74,600 | 1,299,100 | 760,807,300 | 590 |
| 75 | 10,427,000 | 74,600 | 1,299,100 | 1,942,185,200 | 1500 |
| $R_e$ | | | | | |
| 1.1 | 2,640,400 | 6600 | 98,000 | 299,493,000 | 3050 |
| 1.4 (Base case) | 10,427,000 | 74,600 | 1,299,100 | −428,052,700 | Cost-saving |
| 1.8 | 5,955,700 | 110,500 | 1,957,700 | 129,359,500 | 70 |
| Two-wave epidemic[e] | 13,696,300 | 62,700 | 1,072,500 | −682,063,700 | Cost-saving |
| Prior immunity to SARS-CoV-2, % of population | | | | | |
| 10 | 8,025,900 | 147,200 | 2,581,000 | 85,889,700 | 30 |
| 20 | 9,087,700 | 119,000 | 2,168,000 | 55,790,700 | 30 |
| 30 (Base case) | 10,427,000 | 74,600 | 1,299,100 | −428,052,700 | Cost-saving |
| 40 | 7,127,300 | 18,000 | 279,500 | −252,757,900 | Cost-saving |
| 50 | 608,300 | 1500 | 24,300 | 545,399,700 | 22,460 |
| Initial prevalence of active COVID-19, % of population | | | | | |
| 0.05%[f] | 12,247,900 | 70,300 | 1,269,000 | −557,621,500 | Cost-saving |
| 0.1% (Base case) | 10,427,000 | 74,600 | 1,299,100 | −428,052,700 | Cost-saving |
| 0.2% | 8,403,300 | 72,300 | 1,288,700 | −180,874,600 | Cost-saving |
| 0.5% | 6,028,800 | 64,100 | 1,119,800 | 51,633,800 | 50 |

*ICER* incremental cost-effectiveness ratio, $R_e$ effective reproduction number, *USD* United States dollars, *YLS* year-of-life saved.
[a]In these scenario analyses, the reference vaccination program (67% supply, 150,000 vaccinations per day) is compared with no vaccination program under different scenarios. Displayed life-years and costs are rounded to the nearest hundred, whereas ICERs are calculated based on non-rounded life-years and costs, and then rounded to the nearest ten. Cost-saving reflects more years-of-life (greater clinical benefit) and lower costs, and therefore ICERs are not displayed.
[b]In the scenario analysis of a vaccine with 75% effectiveness in preventing SARS-CoV-2 infection, the effectiveness in preventing mild/moderate COVID-19 disease was adjusted to avoid a scenario in which a vaccine has higher effectiveness in preventing infection than it does in preventing symptomatic disease.
[c]Vaccine effectiveness in preventing mild/moderate COVID-19 (apart from severe/critical disease) has minimal impact on the number of deaths. Therefore, seemingly counterintuitive results are due to stochastic variability in the microsimulation. In the analysis of a vaccine that is 29% effective in preventing mild/moderate COVID-19, the vaccine effectiveness in preventing SARS-CoV-2 infection was adjusted to avoid a scenario in which a vaccine is more effective in preventing infection than in preventing symptomatic disease.
[d]Vaccine effectiveness in preventing severe/critical COVID-19 itself has minimal impact on transmission and the number of infections. Therefore, seemingly counterintuitive results are due to stochastic variability in the microsimulation. In the analysis of a vaccine that is 40% effective in preventing severe COVID-19 requiring hospitalization, the vaccine effectiveness in preventing mild/moderate COVID-19 was adjusted to avoid a scenario in which a vaccine is more effective in preventing symptomatic disease than in preventing severe disease requiring hospitalization.
[e]In the analysis of an epidemic with periodic surges, the basic reproduction number ($R_o$) alternates between low and high values over time, and the $R_e$ changes day-to-day as the epidemic and vaccination program progress and there are fewer susceptible individuals. For most of the simulation horizon, $R_o$ is 1.6 (equivalent to an initial $R_e$ of 1.1). However, during days 90–150 and 240–300 of the simulation, $R_o$ is increased to 2.6. This results in two epidemic waves with peak $R_e$ of ~1.4–1.5.
[f]When the initial prevalence of active SARS-CoV-2 infection is 0.05%, the epidemic peak occurs more than 180 days into the simulation. As our modeled time horizon only considers outcomes occurring through day 360, delaying the epidemic peak leads to a small decrease in the number of infections and deaths that are recorded in the scenario without vaccines. As a result, the absolute number of deaths prevented by vaccination decreases slightly as initial prevalence of active infection is changed from 0.1% to 0.05%, even though a greater proportion of deaths are prevented.

(Supplementary Table 3). When several of the main analyses were repeated with lower costs of hospital and ICU care, some ICERs increased, but vaccine supplies of 40% or 80% remained non-dominated (with the latter providing greater clinical benefit), whereas a faster vaccination pace still resulted in greater clinical benefit and lower costs (Supplementary Table 4).

The influence of different scenarios into which the vaccination program would be introduced on cumulative infections, deaths, and health care costs is depicted in Fig. 1. Varying the prevalence of prior immunity and $R_e$ had the greatest influence on both infections and deaths, whereas varying the cost per person vaccinated had the greatest influence on health care costs. Vaccine effectiveness against infection and effectiveness against severe disease requiring hospitalization were more influential than effectiveness against mild/moderate disease in terms of reductions in deaths and costs.

**Multi-way sensitivity analyses**. In a multi-way sensitivity analysis in which we simultaneously varied vaccine effectiveness against

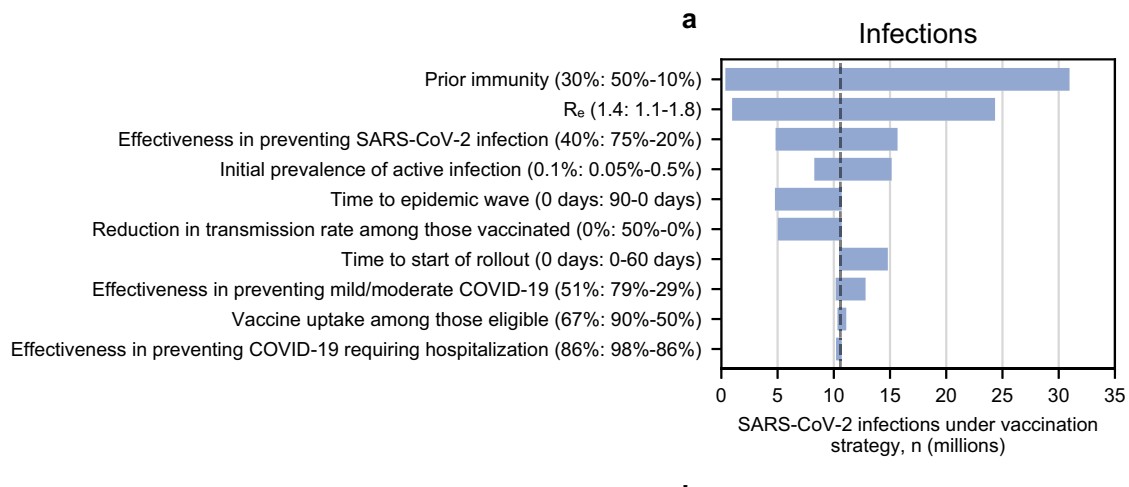

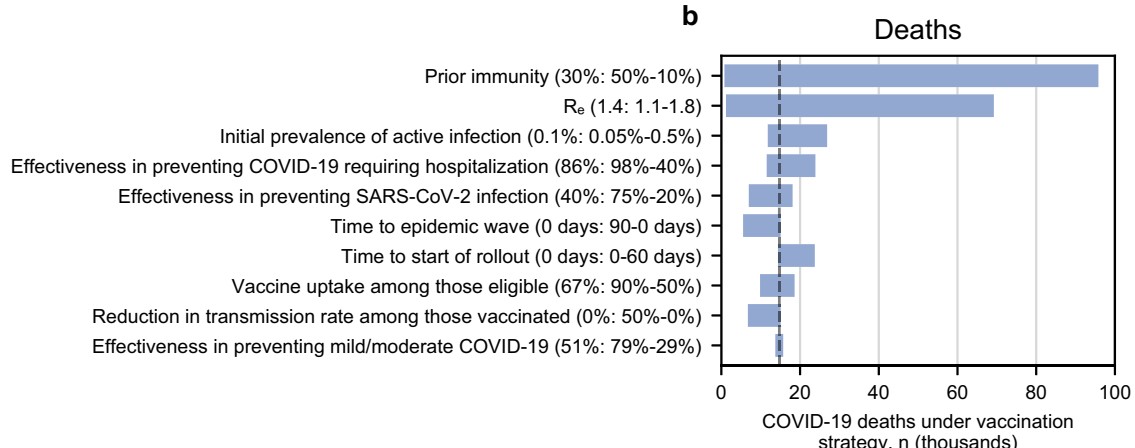

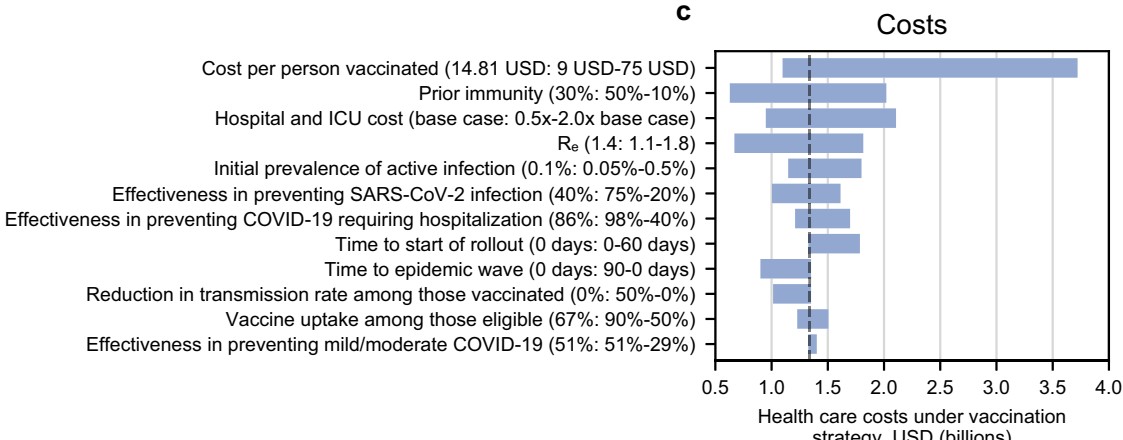

**Fig. 1 One-way sensitivity analysis: influence of each parameter on cumulative SARS-CoV-2 infections, COVID-19 deaths, and health care costs.** This tornado diagram demonstrates the relative influence of varying each key model parameter on clinical and economic outcomes over 360 days. This is intended to reflect the different scenarios in which a reference vaccination program (vaccine supply sufficient for 67% of South Africa's population, pace 150,000 vaccinations per day) might be implemented. The dashed line represents the base case scenario for each parameter. Each parameter is listed on the vertical axis and in parentheses are the base case value and, after a colon, the range examined. The number on the left of the range represents the left-most part of the corresponding bar and the number on the right of the range represents the right-most part of the corresponding bar. The horizontal axis shows the following outcomes of a reference vaccination program: **a** cumulative SARS-CoV-2 infections; **b** cumulative COVID-19 deaths; **c** cumulative health care costs. In some analyses, the lowest or highest value of an examined parameter produced a result that fell in the middle of the displayed range of results, due to stochastic variability when the range of results was narrow.

infection and cost per person vaccinated, the reference vaccination program was cost-saving compared with a scenario without vaccines when cost per person vaccinated was $14.81, even when effectiveness against infection was as low as 20% (Fig. 2). When

cost per person vaccinated was $25, the program was cost-saving when effectiveness against infection was at least 40%. Even at the highest examined cost per person vaccinated ($75) and the lowest examined effectiveness against infection (20%), the vaccination

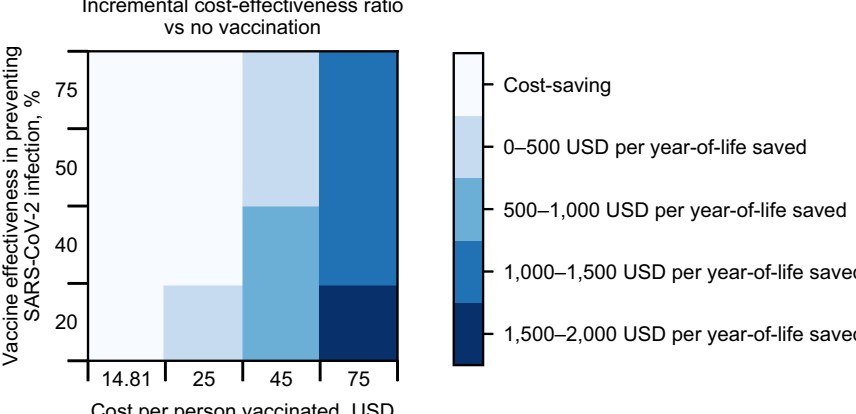

**Fig. 2 Multi-way sensitivity analysis of vaccine effectiveness against infection and vaccination cost: incremental cost-effectiveness ratio of vaccination program compared with no vaccination.** Each box in the 4 × 4 plot is colored according to the incremental cost-effectiveness ratio (ICER). The lightest color represents scenarios in which a reference vaccination program (vaccine supply sufficient for 67% of South Africa's population, pace 150,000 vaccinations per day) is cost-saving compared with no vaccination program, meaning that it results in clinical benefit and reduces overall health care costs. The darker colors reflect increasing ICERs, whereby a reference vaccination program, compared with no vaccination program, results in both clinical benefit and higher overall health care costs. The ICER is the model-generated difference in costs divided by the difference in years-of-life between a reference vaccination program and no vaccination program. In none of these scenarios is the ICER above $2000/year-of-life saved (YLS).

program had an ICER < $2000/YLS compared with no vaccination program (Fig. 2).

We performed several additional multi-way sensitivity analyses in which we simultaneously varied combinations of vaccine supply, vaccination pace, vaccine effectiveness against infection, cost per person vaccinated, $R_e$, and prevalence of prior immunity (Table 3 and Supplementary Figs. 4–8). Of note, to optimize efficiency, increasing vaccination pace was more important than increasing vaccine supply. At a cost of $45 or $75 per person vaccinated, increasing vaccination pace led to similar or lower ICER (greater economic efficiency), while increasing vaccine supply led to a similar or higher ICER (less economic efficiency) (Supplementary Fig. 4). At a cost up to $25 per person vaccinated, the vaccination program was cost-saving under nearly all strategies and scenarios (Supplementary Figs. 4–6). Even when the vaccination program increased costs, the ICERs were <$2000/YLS compared with a scenario without vaccines (Supplementary Figs. 4–6).

## Discussion

Using a dynamic COVID-19 microsimulation model, we found that vaccinating 67% of South Africa's population, meeting the government's goal for 2021[16], would both decrease COVID-19 deaths and reduce overall health care costs compared with a scenario without vaccines or with a 20% vaccine supply, by reducing the number of infections, hospitalizations, and ICU admissions. Further increasing the vaccine supply to 80%, while simultaneously increasing vaccine acceptance, would save even more lives while modestly increasing costs. Vaccination pace—the number of vaccine doses administered daily--rather than supply itself may be most influential to maximizing public health benefits and economic efficiency. Increasing the pace would reduce both deaths and overall health care costs. The program remained cost-saving even with conservative estimates of vaccine effectiveness and with higher per-person vaccination costs, highlighting that the characteristics of vaccination program implementation are likely to be more influential than the characteristics of the vaccine itself. Furthermore, the vaccination program remained economically efficient (either cost-saving or with a relatively low ICER representing good clinical value for additional money spent) across most epidemic scenarios,

including various rates of epidemic growth and a broad range of prevalence of prior population immunity. Although there is no consensus on an ICER threshold for cost-effectiveness in South Africa, for context, the country's gross domestic product per capita in 2019 was ~$6000 and a published South Africa cost-effectiveness threshold from an opportunity cost approach was ~$2950 (2020 USDs) per disability-adjusted life-year averted[19,20].

Much has been made about differences in the leading vaccine candidates and the impact of variants, such as the B.1.351 (β) variant, which eventually accounted for over 90% of SARS-CoV-2 infections in South Africa and the B.1.617.2 (δ) variant, on vaccine effectiveness[4,15]. However, we found that, even with substantially lower vaccine efficacy than reported in clinical trials, vaccination programs would prevent the majority of COVID-19 deaths compared to scenarios without vaccines. For example, decreasing vaccine effectiveness against mild/moderate disease and severe/critical disease requiring hospitalization to 40% still reduced COVID-19 deaths by 65,800 (74%) compared with a scenario without vaccines. Although efficacy against symptomatic and severe disease have been the focus of vaccine trials, these parameters were less influential on population-wide health and cost outcomes than efficacy against infection, which is less commonly reported in trials[1–4]. Nonetheless, the effectiveness ranges we examined in sensitivity analysis include the point estimates of efficacy against symptomatic and severe disease reported in clinical trials of the AstraZeneca ChAdOx1, Moderna mRNA-1273, and Pfizer-BioNTech mRNA BNT162b2 vaccines[1–3]. This suggests that all of these vaccines are likely to have both health and economic benefits. Furthermore, our sensitivity analysis examining different $R_e$ scenarios likely captures the potential influence of more contagious SARS-CoV-2 variants such as δ.

Similarly, we found that vaccination programs remained economically favorable even at relatively high vaccination costs. Although we did not explicitly account for all implementation and scale-up costs of a vaccination program, our estimates of cost per person vaccinated were based on reported costs of both vaccine and delivery in South Africa[21–23]. Achieving the government's goal of vaccinating 67% of South Africans within 1 year will depend at least partially on global vaccine supplies and may require global policymakers to better fund and facilitate vaccine distribution and accessible pricing for LMICs, in addition to local

attention to delivery infrastructure and community outreach. Although these expenses may increase program costs, we found that the vaccination program would remain cost-saving at a vaccination cost up to $25/person and likely cost-effective even at per-person vaccination cost up to $75/person (ICER $1500/YLS). This is due to cost offsets in preventing hospitalizations.

A faster pace of vaccination consistently decreased infections, deaths, and costs across a range of epidemic growth scenarios. Yet, this was not always true of a higher vaccine supply. With lower epidemic growth ($R_e = 1.1$), which approximates the basic reproduction number in the intra-wave periods in South Africa, a faster pace remained preferable from a clinical and economic standpoint. However, with the faster vaccination pace, increasing the proportion of the population vaccinated from 40% to 67% resulted in higher costs and only modestly fewer YLL, with an ICER of $85,290/YLS, well above commonly reported willingness-to-pay thresholds in South Africa[20,24–27]. By contrast, when a higher epidemic growth rate is seen ($R_e = 1.8$), as was documented during the first and second waves in South Africa, a faster vaccination pace remained highly preferable, and increasing the proportion of the population vaccinated from 40% to 67% resulted in fewer YLL and higher costs with a much lower ICER of $1160/YLS. Overall, these results demonstrate the importance of rolling out vaccinations quickly, particularly ahead of any future waves of the epidemic. Consequently, policymakers should invest in establishing a vaccine distribution and administration system, to ensure vaccines will be administered as promptly as possible. All available distribution channels, including those in public and private sectors, should be leveraged.

Our model projections were sensitive to $R_e$ and to the prevalence of prior immunity to SARS-CoV-2. However, vaccination was generally economically efficient even in scenarios of very low epidemic growth, albeit in some instances with a lower supply target. When the prevalence of prior protective immunity was increased to 50%, the ICER rose substantially. We assumed that prior infection protects against another SARS-CoV-2 infection for the duration of the simulation period. If this is not the case, either because immunity wanes or viral variants make prior infection poorly protective against re-infection, as appeared to be seen in the second waves in South Africa and Brazil, then the vaccination program could still provide good value even with a high prevalence of prior infection[28,29].

These results should be interpreted within the context of several limitations. We assumed that vaccine effectiveness was constant starting 14 days after administration and continuing throughout the 360-day simulation. Early data suggest that post-vaccination immunity lasts at least for months[1–3,30,31]. Our model assumes homogeneous mixing of the entire population. This assumption may result in conservative estimates of cost-effectiveness of vaccination, in particular at lower supply levels, because herd immunity is likely to be achieved at lower rates of vaccination after accounting for heterogeneous mixing[32]. There may be economies of scale such that the cost per person vaccinated decreases as the vaccine supply or vaccination pace increase and vaccination program resources are used more efficiently. Our modeled vaccination prioritization was based exclusively on age and not on employment type, comorbidity presence, or urban/rural heterogeneity in epidemiology or vaccination delivery. Vaccination programs that reach vulnerable and disadvantaged groups would likely improve population-level health outcomes and health equity. Long-term disability among some of those who recover from COVID-19 is an important consideration for policymakers not captured by our model, which considers only YLL due to premature mortality. Our vaccination cost-effectiveness results may therefore be conservative, in particular among younger age groups that are less likely to die from COVID-19 but are still at risk for long-term sequelae[33]. We did not consider the impact of

COVID-19 or vaccination on other health care programs (e.g., HIV and tuberculosis care). We assessed costs from a health care sector perspective and did not account for other sector costs associated with lockdowns and failure to achieve epidemic suppression (e.g., macroeconomic factors such as job and productivity losses, and microeconomic factors such as reduced household income and disruptions to education)[34,35]. Excluding these costs may underestimate the true value of COVID-19 vaccination to society. We did not explicitly model the use of non-pharmaceutical interventions (NPIs) as a standalone strategy or in combination with vaccination. However, the evaluation of various transmission scenarios (including a sensitivity analysis in which $R_0$ changes over time) captures the potential impacts of different levels of NPI implementation on clinical outcomes. As with all modeling exercises, our results are contingent on assumptions and input parameters. Primary assumptions in our model included initial prevalence of COVID-19, prevalence of prior immunity, time to vaccine rollout, and vaccine efficacy against asymptomatic infection.

Given data limitations and the uncertainty in making long-term projections, we limited the time horizon of our analysis to 1 year. The sustainability and cost-effectiveness of vaccination beyond 1 year is likely to depend on the duration of protection conferred by existing vaccines, their effectiveness against emergent variants, and the costs, effectiveness, and frequency of potential booster shots—factors that remain unknown as of June 2021. If SARS-CoV-2 becomes endemic, cost-effectiveness analysis will become increasingly critical for integrating vaccination programs within health program budgets.

In summary, we found that a COVID-19 vaccination program would reduce infections and deaths, and likely reduce overall health care costs in South Africa across a range of possible scenarios, even with conservative assumptions around vaccine effectiveness. Our model simulations underscore that in South Africa and similar settings, acquisition and rapid distribution of vaccines should be prioritized over relatively small differences in vaccine effectiveness and price. The pace of vaccination is as or more important than population coverage and, therefore, attention to vaccination program infrastructure is critical. Non-pharmaceutical practices such as mask wearing and physical distancing remain crucial to reduce epidemic growth, while vaccination programs are being implemented[10]. Policymakers can use our results to guide decisions about vaccine selection, supply, and distribution to maximally reduce the deleterious impact of the COVID-19 pandemic in South Africa.

## Methods

**Analytic overview.** We used the Clinical and Economic Analysis of COVID-19 Interventions (CEACOV) dynamic state-transition Monte Carlo microsimulation model to reflect COVID-19 natural history, vaccination, and treatment[36]. We previously used the CEACOV model to project COVID-19 clinical and economic outcomes in a variety of settings, including an analysis of non-pharmaceutical public health interventions in South Africa[24,37–39].

Starting with SARS-CoV-2 active infection prevalence of 0.1% (or ~60,000 active cases, roughly 10 times the number reported in the first 10 days of April 2021), we projected clinical and economic outcomes over 360 days, including daily and cumulative infections, deaths, hospital and ICU bed use, and health care costs without discounting[40]. Outside the model, we calculated the mean lifetime YLS from each averted COVID-19 death during the 360-day model horizon, stratified by age (mean 17.77 YLS per averted COVID-19 death across all individuals, Supplementary Methods). We did not include costs beyond the 360-day model horizon[24]. We determined the ICER, the difference in health care costs (2020 USDs) divided by the difference in life-years between different vaccination strategies. Our ICER estimates include health care costs during the 360-day model horizon and YLS over a lifetime from averted COVID-19 deaths during the 360-day model horizon[24]. "Cost-saving" strategies were those resulting in higher clinical benefits (fewer life-years lost) and lower costs than an alternative.

**Model structure.** In each simulation, we assumed a fixed supply of vaccines that would be administered to eligible and willing individuals regardless of history of

SARS-CoV-2 infection. Available vaccine doses would first be offered to those aged ≥60 years, then to those aged 20–59 years, and finally to those aged <20 years[41].

In the base case, we applied characteristics of Ad26.COV2.S (Johnson & Johnson/Janssen), a single-dose vaccine for which administration in South Africa began through a phase 3b study in health care workers in February 2021[4,42]. To reflect possible implementation of other vaccines, as well as published data and uncertainties around the type of protection provided by each vaccine, we varied vaccine effectiveness against SARS-CoV-2 infection, effectiveness against mild/moderate COVID-19 disease, and effectiveness against severe COVID-19 disease requiring hospitalization. We assumed that a single vaccine dose would be given and did not explicitly model a two-dose schedule.

At model initiation, each individual is either susceptible to SARS-CoV-2, infected with SARS-CoV-2, or immune (by way of prior infection). Each susceptible individual faces a daily probability of SARS-CoV-2 infection. Once infected, an individual moves to the pre-infectious latency state and faces age-dependent probabilities of developing asymptomatic, mild/moderate, severe, or critical disease (Supplementary Methods, Supplementary Table 5, and Supplementary Fig. 1). Individuals with severe or critical disease are referred to hospitals and ICUs, respectively. If hospital/ICU bed capacity has been reached, the individual receives the next lower available intervention, which is associated with different mortality risk and cost (e.g., if a person needs ICU care when no ICU beds are available, they receive non-ICU hospital care). Details of COVID-19 transmission, natural history, and hospital care in the model are described elsewhere and in the Supplementary Methods[24].

**Input parameters.** We defined the age distribution based on 2019 South Africa population estimates, in which 37% were aged <20 years, 54% were 20–59 years,

and 9% were ≥60 years (Table 3)[43]. We assumed in the base case that, at model initiation, 30% had prior infection and were immune to repeat infection. This assumption was based on an estimate of the proportion of South Africa's population that had been exposed to the B.1.351 variant by 30 January 2021 (Supplementary Methods)[15,44–46].

In the reference vaccination program strategy, we assumed the following: (a) there would be a sufficient supply of vaccine doses to fully vaccinate 67% of South Africa's population (~40 million vaccinated people)[16]; (b) pace of vaccination was 150,000 doses/day[17,18]. Our comparisons of different vaccination program strategies included varying the vaccine supply to that sufficient to cover 0–80% of South Africa's population and increasing the pace of vaccination up to 300,000 doses/day. In the base case, we assumed that vaccine uptake among those eligible was 67%, accounting for vaccine hesitancy and failure to reach some individuals[47,48]. Vaccine effectiveness was 40% against infection, 51% against mild/moderate disease, and 86% against severe or critical disease requiring hospitalization. The latter two were based on reported efficacies of the Johnson & Johnson/Janssen vaccine ≥14 days post vaccination in South Africa[4].

Supplementary Table 5 indicates daily disease progression probabilities, age-dependent probabilities of developing severe or critical disease, and age-dependent mortality probabilities for those with critical disease. We stratified transmission rates by disease state, adjusting them to reflect an initial effective reproduction number ($R_e$) = 1.4 in the base case[49]. We also simulated alternative epidemic growth scenarios with lower or higher initial $R_e$ and a scenario in which there were episodic surges above a lower background basic reproduction number ($R_0$), as observed in the South Africa epidemic over the past year (Supplementary Methods).

The maximum availability of hospital and ICU beds per day was 119,400 and 3300, respectively (Table 3)[50]. We applied vaccination costs and daily costs of hospital care and ICU care based on published estimates and/or cost quotes

**Table 3 Input parameters for a model-based analysis of COVID-19 vaccination in South Africa.**

| Parameter | Base case value (range) | Sources |
|---|---|---|
| Initial state | | |
| Age distribution, % | | 43 |
| <20 Years | 37 | |
| 20–59 Years | 54 | |
| ≥60 Years | 9 | |
| Initial health state distribution, % | | |
| Susceptible | 69.9 (49.9–89.9) | Assumption |
| Infected with SARS-CoV-2 | 0.1 (0.05–0.5) | Assumption[a] |
| Recovered (prior immunity) | 30 (10–50) | 15, 44–46 |
| Transmission dynamics | | |
| Effective reproduction number, $R_e$ | 1.4 (1.1–1.8) | 49 |
| Time to start of epidemic wave, days | 0 (0–90) | Assumption |
| Relative reduction in onward transmission rate among vaccinated individuals, % | 0 (0–50) | Assumption |
| Hospital and ICU care | | |
| Resource availabilities | | |
| Hospital beds, daily, n | 119,400 | 50 |
| ICU beds, daily, n | 3300 | 50 |
| Costs | | |
| Hospitalization, daily, USD | 154 (77–309) | 52–55 |
| ICU care[b], daily, USD | 1751 (798–3502) | 53–56 |
| Vaccination program strategies | | |
| Vaccine supply, % of population | 67 (20–80) | 16 |
| Vaccinations per day, n | 150,000 (150,000–300,000) | 17, 18 |
| Time to rollout start, days | 0 (0–60) | Assumption |
| Vaccine characteristics[c] | | |
| Effectiveness in preventing SARS-CoV-2 infection, % | 40 (20–75) | Assumption |
| Effectiveness in preventing mild/moderate COVID-19 disease[d], % | 51 (29–79) | Age-dependent assumptions[4], |
| Effectiveness in preventing severe or critical COVID-19 disease requiring hospitalization, % | 86 (40–98) | 4 |
| Number of doses required for effectiveness | 1 | 4 |
| Time to effectiveness, days post vaccination | 14 | 4 |
| Vaccine uptake among those eligible, % | 67 (50–90) | 48 |
| Vaccination cost per person, USD | 14.81 (9–75) | 21–23, 54, 55 |

Ranges reflect values examined in analyses of alternative vaccination program strategies and in sensitivity analyses of different vaccine characteristics and epidemic growth scenarios.
ICU intensive care unit, $R_e$ effective reproduction number, USD United States dollars.
[a]Initial prevalence of each state of infection and disease are in Supplementary Table 5.
[b]The range of ICU care costs includes the cost (from Edoka et al.[53]) applied in a repeat of several of the main analyses.
[c]In the base case, we model a vaccination program based on characteristics of the Johnson & Johnson/Janssen Ad26.COV2.S vaccine[4]. In sensitivity analyses, vaccine effectiveness and cost are varied across a range of possible values, to evaluate the influence of these parameters on clinical and economic outcomes and to account for uncertainty around published estimates.
[d]Values reflect the weighted average of vaccine effectiveness in preventing mild/moderate COVID-19 across age groups. Our modeled vaccine effectiveness in preventing mild/moderate COVID-19 was specified in an age-dependent manner to reflect the reported efficacy of the Ad26.COV2.S vaccine in preventing moderate to severe/critical COVID-19 in South Africa[4]. In the base case, this results in 52% effectiveness in preventing any symptomatic COVID-19 across all age groups. In sensitivity analysis, this value is varied from 30% to 79%.

obtained in South Africa (Table 3 and Supplementary Methods). In the base case, we applied a total vaccination cost of $14.81 per person, based on estimated costs in South Africa of $10/dose for the vaccine and $4.81/dose for service and delivery (Supplementary Methods)[21–23]. We varied vaccination costs in sensitivity analyses.

**Validation**. We previously validated our natural history assumptions by comparing model-projected COVID-19 deaths with those reported in South Africa[24]. We updated our validation by comparing the model-projected number of COVID-19 infections and deaths with the number of cases and deaths reported in South Africa through 10 April 2021, accounting for underreporting (Supplementary Methods and Supplementary Fig. 3)[40,51].

**Sensitivity analysis**. We used sensitivity analysis to examine the relative influence on clinical and cost projections of various parameters around vaccine characteristics and epidemic growth (Table 3). Specifically, we varied the following: vaccine acceptance (50–90% among eligible individuals); vaccine effectiveness in preventing infection (20–75%), mild/moderate disease (29–79%), and severe/critical disease requiring hospitalization (40–98%); cost ($9–75/person); initial prevalence of COVID-19 disease (0.05–0.5%); initial $R_e$ (1.1–1.8); prior immunity (10–50% of population); reduction in transmission rate among vaccinated but infected individuals (0–50%); and hospital and ICU daily costs (0.5×–2.0× base case costs). The ranges of vaccine effectiveness against mild/moderate disease and severe/critical disease requiring hospitalization were based on efficacies and 95% confidence intervals reported in the Johnson & Johnson/Janssen vaccine trial (Supplementary Methods)[4]. We also examined ICERs when the relatively high costs of ICU care were excluded and when all hospital care costs (non-ICU and ICU) were excluded. We performed multi-way sensitivity analyses in which we simultaneously varied parameters influential in one-way sensitivity analyses.

**Reporting summary**. Further information on research design is available in the Nature Research Reporting Summary linked to this article.

## Data availability
This modeling study involved the use of published or publicly available data. The data used and the sources are described in the manuscript and Supplementary Information. No primary data were collected for this study. Model flowcharts are in the Supplementary Information.

## Code availability
The simulation model code is available at https://zenodo.org/record/5565320 (https://doi.org/10.5281/zenodo.5565320).

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

## Acknowledgements

This research was supported by the National Institute of Allergy and Infectious Diseases of the National Institutes of Health [R37 AI058736-16S1] (K.A.F.) and by the Wellcome Trust [Grant number 210479/Z/18/Z] (G.H.). For the purpose of open access, we applied a CC BY public copyright license to any Author Accepted Manuscript version arising from this submission. The funding sources had no role in the study design, data collection, data analysis, data interpretation, writing of the manuscript, or in the decision to submit the manuscript for publication. The content is solely the responsibility of the authors and does not necessarily represent the official views of the funding sources. We thank Nattanicha Wattananimitgul, Eli Schwamm, and Nora Mulroy for technical assistance.

## Author contributions

All authors contributed substantively to this manuscript in the following ways. Study and model design: K.P.R., K.P.F., J.A.S., G.H., R.J.L., C.P., F.M.S., K.A.F., and M.J.S. Data analysis: K.P.R., K.P.F., J.A.S., F.M.S., K.A.F., and M.J.S. Interpretation of results: K.P.R., K.P.F., J.A.S., G.H., R.J.L., C.P., F.M.S., K.A.F., and M.J.S. Drafting the manuscript: K.P.R. and M.J.S. Critical revision of the manuscript: K.P.R., K.P.F., J.A.S., G.H., R.J.L., C.P., F.M.S., K.A.F., and M.J.S. Final approval of the submitted version: K.P.R., K.P.F., J.A.S., G.H., R.J.L., C.P., F.M.S., K.A.F., and M.J.S..

## Competing interests

R.J.L. serves on South Africa's Ministerial Advisory Committee on COVID-19 Vaccines (VMAC). The authors declare no additional competing interests.
