## [Peer Review File · Nature Communications]

REVIEWER COMMENTS

Reviewer #1 (Remarks to the Author):

Thank you for the opportunity to review this manuscript. In this study, the authors present a previously validated model for SARS-CoV-2 transmission and COVID-19 disease, for the setting of South Africa. They simulate vaccine rollout relative to an ongoing epidemic trajectory, and assess the benefit of vaccination in terms of health benefit, and incremental cost effectiveness, for a range of epidemic characteristics, vaccine properties, and vaccine delivery scenarios (including pace of vaccination). They found that a vaccine would be beneficial and cost-effective, and that even for a moderately efficacious vaccine, it would still be cost-effective. The rollout strategy was found to be as or more important compared to the properties of the vaccine product. I found this manuscript to be very well-written, and it clearly described the model formulation, assumptions, and findings. The results tables and figures were nicely presented and readily interpretable. Cost-effectiveness of COVID-19 vaccination as not yet been widely explored due to still being in the “pandemic” phase, and with the focus being more on how to best distribute doses globally and ensure equitable access in resource-limited settings. However, the topic of this manuscript is important because if we reach a stage in the coming years where COVID-19 becomes endemic, cost-effectiveness considerations will be critical for integrating SARS-CoV-2 vaccination within ongoing country-specific vaccination programs. I believe these results would be of interest to a wide audience.

My comments are mostly related to areas where additional clarification could be provided, and a question about vaccine efficacy calculations.

Major comments

1. I note that you described in the Supplementary Material the previous validation of your model that indicated a than around 10% of infections are reported as cases, and that this then informed your starting prevalence in your simulation of 0.1% (10 times the number of cases reported in the first 10 days of April 2021). However, do you think it is possible that the level of underreporting of cases could change over time? Did you consider testing sensitivity to this 0.1% starting prevalence? As this would then influence the ongoing epidemic trajectory upon which your results are highly sensitive to. If it is not feasible to test this sensitivity, I think it is important to provide some discussion of how robust you think this estimate is and what influence it has on the outcomes of the analysis.

2. This partly relates to my first comment. It is clear that the results you present are highly sensitive to epidemic history and projected epidemic going forward in time (this is a difficulty for all COVID-19 modelling studies!). E.g. you note that “Varying the prevalence of prior immunity and R_e had the greatest influence on both infections and deaths”. I do think that the way you conceptualised the prior and future epidemic makes logical sense, but I had trouble visualising the dynamics of the epidemic patterns. I think that you need to be much clearer upfront (including in the abstract) about the sensitivity of results to assumptions about the prior and future epidemic. It would also be of benefit to the reader for these counterfactual or baseline epidemic trajectories (without vaccination) for different levels of R_e to be plotted over time (perhaps in the Supplement as you had done for Figure S8). Otherwise, it is difficult to really understand what the different assumptions about R_e going forward mean for the modelled size and duration of the epidemic wave.

3. Line 180. Cohort characteristics. Did you assume a further cohort structure within these broad age bands? What did you assume about contact patterns between age or risk groups?

4. Pace of vaccination. It makes sense that the pace of vaccination is important (more people

vaccinated and therefore protected within a particular time period) but I think that it really needs to be emphasised that what this really shows is the number of people that can be vaccinated relative to the next COVID-19 epidemic wave – i.e. what is assumed about pace is highly dependent on assumptions about future transmission.

5. Supplementary Material, page 9 (vaccine efficacy calculations). I did not interpret different endpoints of vaccine efficacy as being additive in this way. E.g. If vaccine efficacy against infection is 40%, and efficacy against disease is 60%, then the efficacy against disease in individuals who still become infected (your measure of P(B) as I understood it) would be calculated as $1 - (1 - \text{efficacy_disease}) / (1 - \text{efficacy_infection})$. Could you please clarify your reasoning for these equations?

Minor comments.

Abstract, line 68. Should this be “providing a vaccine” or “providing vaccination”.

Line 196. Should this be “some individuals”?

Results. First line – it would be helpful to restate here the analysis period over which you describe these results.

Line 332. “However, we found that even with lower effectiveness than that estimated from studies, a vaccination program would still prevent the majority of COVID-19 deaths that would occur without a vaccination program.” I am a bit unclear on what you mean in this sentence – when you say “majority of” it sounds like deaths are being prevented some other way? Perhaps consider rephrasing.

Line 389, discussion section. Good to just briefly state here again what the uncertain parameters were?

Line 394. I think you mean to refer to “model simulations” here (or similar) – rather than “data”?

Reviewer #2 (Remarks to the Author):

Major claim of the paper is that optimal COVID-19 vaccination program would reduce deaths and health care costs in South Africa. Implementation factors including procurement, distribution and pace of vaccine roll out are more influential than characteristics of the vaccine itself such as vaccine efficacy and price in maximizing public health benefits and economic efficiency.

The concept proposed and resulting interpretation is novel and of interest to others in the wider community.

However, in terms of the conceptual framework and assumptions I have the following major comments.

First, the interpretation of the results given wide confidence intervals e.g. around R values etc should be presented with more care. Claiming 40% vaccination coverage is an optimal vaccination strategy in a South African setting can be harmful. The results have lots of uncertainties that heavily rely on assumptions with wide confidence intervals (vaccine efficacy while having variants of concern circulating, optimistic vaccine procurement prices etc) and based on a narrow scope of analyses. There is no mentioning of equity concerns among different target groups in the analyses. Also, vaccination

strategies can be incrementally cost-effective but the decision makers in the current pandemic face broader questions, specifically on the health-non-health e.g. macroeconomic trade-offs. The incremental CE analyses does not take these issues into account. In the discussion the authors should highlight these limitations when interpreting the results for policy recommendations.

Second, related to the previous point. Disability is not included in the analyses which makes not only the \$/DALY threshold problematic but also not taking into account the impact of COVID-19 vaccines to younger/working populations relevant in a demographic setting like SA. The authors could refer to a recent paper that came out in Nature by Briggs and Vassall (see: <https://www.nature.com/articles/d41586-021-01392-2>)

Third, also related to the long term effects of COVID why only looking at 1 year? What about vaccine effectiveness and duration protection? What about the sustainability of programs beyond one year? Would result look different beyond one year when public health and social measures (PHSM) are lifted? Is it realistic in SA to reach coverage rates of over 60% in one year? What about the cost function to reaching difficult to reach populations?

Fourth, it is not clear what the authors assume in both the no vaccination and vaccination strategies with regards to other PHSM that are in place? Will these be lifted or remain the same when vaccination strategies are introduced? How will these influence the results.

Finally a couple of minor points.

The authors claim that vaccine supply (and pace) can exert more direct control by policy makers (line 80 in supplement). Not sure this is true given the global vaccine supply shortage, specifically for LMICs. Also supply and access to vaccines also depends indirectly on procurement price of vaccines.

Delivery costs from influenza focusing on elderly, PLWHA, ULMC etc are higher (5.42-8,22) to according to Edoka (2021) compared to the vaccine delivery costs per person (line 254 in supplement).

RESPONSES TO REVIEWER COMMENTS

Page numbers refer to the tracked versions of the revised manuscript and supplement.

Reviewer #1 (Remarks to the Author):

Thank you for the opportunity to review this manuscript. In this study, the authors present a previously validated model for SARS-CoV-2 transmission and COVID-19 disease, for the setting of South Africa. They simulate vaccine rollout relative to an ongoing epidemic trajectory, and assess the benefit of vaccination in terms of health benefit, and incremental cost effectiveness, for a range of epidemic characteristics, vaccine properties, and vaccine delivery scenarios (including pace of vaccination). They found that a vaccine would be beneficial and cost-effective, and that even for a moderately efficacious vaccine, it would still be cost-effective. The rollout strategy was found to be as or more important compared to the properties of the vaccine product. I found this manuscript to be very well-written, and it clearly described the model formulation, assumptions, and findings. The results tables and figures were nicely presented and readily interpretable. Cost-effectiveness of COVID-19 vaccination as not yet been widely explored due to still being in the “pandemic” phase, and with the focus being more on how to best distribute doses globally and ensure equitable access in resource-limited settings. However, the topic of this manuscript is important because if we reach a stage in the coming years where COVID-19 becomes endemic, cost-effectiveness considerations will be critical for integrating SARS-CoV-2 vaccination within ongoing country-specific vaccination programs. I believe these results would be of interest to a wide audience.

Response: We thank the reviewer for their thorough consideration of our work and positive feedback.

My comments are mostly related to areas where additional clarification could be provided, and a question about vaccine efficacy calculations.

Major comments

1. I note that you described in the Supplementary Material the previous validation of your model that indicated a than around 10% of infections are reported as cases, and that this then informed your starting prevalence in your simulation of 0.1% (10 times the number of cases reported in the first 10 days of April 2021). However, do you think it is possible that the level of underreporting of cases could change over time? Did you consider testing sensitivity to this 0.1% starting prevalence? As this would then influence the ongoing epidemic trajectory upon which your results are highly sensitive to. If it is not feasible to test this sensitivity, I think it is important to provide some discussion of how robust you think this estimate is and what influence it has on the outcomes of the analysis.

Response: We agree with the reviewer that case reporting is variable and conditional on availability of testing, reporting of test results, test validity, and health care seeking behaviors. Changes in any of these factors could either increase or decrease the ratio of reported cases to total cases. As suggested, we have added sensitivity analyses in which we vary the initial COVID-19 active disease prevalence from as low as 0.05% up to 0.5%. We have added this to Methods, Table 3, and Results.

Methods, pages 20-21: “We used sensitivity analysis to examine the relative influence on clinical and cost projections of various parameters around vaccine characteristics and epidemic growth (Table 3). Specifically, we varied: vaccine acceptance (50-90% among eligible individuals);

vaccine effectiveness in preventing infection (20-75%), mild/moderate disease (29-79%), and severe/critical disease requiring hospitalization (40-98%); cost (\$9-75/person); **initial prevalence of COVID-19 disease (0.05-0.5%)**; initial R_e (1.1-1.8); prior immunity (10-50% of population); reduction in transmission rate among vaccinated but infected individuals (0-50%); and hospital and ICU daily costs (0.5x-2.0x base case costs).”

In this sensitivity analysis, we found that a higher initial prevalence of active COVID-19 reduced the absolute number of infections and deaths prevented by a vaccination program over 360 days because the epidemic peak occurs earlier (when population coverage is still low). However, the vaccination program would remain cost-saving or cost-effective compared to no vaccination even when initial prevalence of active COVID-19 is as high as 0.5%. We have added this to the Results and Table 2.

Results, page 8: “The reference vaccination program had an ICER $< \$100/\text{YLS}$ or was cost-saving compared with a scenario without vaccines across different values of prior immunity (up to 40%), **initial prevalence of active COVID-19**, reduction in transmission rate among vaccinated but infected individuals, and costs of hospital and ICU care (Table 2, Supplementary Table 3).”

2. This partly relates to my first comment. It is clear that the results you present are highly sensitive to epidemic history and projected epidemic going forward in time (this is a difficulty for all COVID-19 modelling studies!). E.g. you note that “Varying the prevalence of prior immunity and R_e had the greatest influence on both infections and deaths”. I do think that the way you conceptualised the prior and future epidemic makes logical sense, but I had trouble visualising the dynamics of the epidemic patterns. I think that you need to be much clearer upfront (including in the abstract) about the sensitivity of results to assumptions about the prior and future epidemic. It would also be of benefit to the reader for these counterfactual or baseline epidemic trajectories (without vaccination) for different levels of R_e to be plotted over time (perhaps in the Supplement as you had done for Figure S8). Otherwise, it is difficult to really understand what the different assumptions about R_e going forward mean for the modelled size and duration of the epidemic wave.

Response: We thank the reviewer for these comments.

As requested, we have added Supplementary Figure 11 to demonstrate the epidemic size over time, in terms of daily and cumulative infections in the absence of vaccination, for each of our epidemic growth scenarios.

Supplementary Figure 11. Modeled scenarios of epidemic growth in the absence of vaccines.

We completely agree that results regarding cost-effectiveness varied, such that less severe epidemics made vaccination relatively less cost-effective and more severe epidemics made vaccination more cost-effective. On the other hand, we believe our results are largely robust to these assumptions. For example, even with a slow-moving epidemic ($R_e=1.1$) the 40% vaccine supply scenario, with a pace of 300,000 vaccinations/day, resulted in an incremental cost-effectiveness ratio (ICER) of \$360/year-of-life saved (YLS) compared with no vaccination. In other words, we see the main added value of our work not in identifying the “goal” supply target *per se*, but in highlighting the cost-effectiveness of various forms of a vaccination program overall and in identifying that the features that maximize a vaccination program’s value can be influenced by policymakers, e.g., prioritizing vaccination pace rate over modest gains in vaccine efficacy or modest changes in vaccine unit costs. We have added text to the Abstract and the Discussion to clarify our goals and contextualize the findings in light of these goals.

Abstract, page 3: **“Model results were most sensitive to assumptions about epidemic growth and prevalence of prior immunity to SARS-CoV-2, though the vaccination program still provided high value and decreased both deaths and health care costs across a wide range of assumptions.”**

Discussion, page 13: **“Our model projections were sensitive to R_e and to the prevalence of prior immunity to SARS-CoV-2. However, vaccination was generally economically efficient **even in scenarios of very low epidemic growth, albeit in some instances with a lower supply target.**”**

When the prevalence of prior protective immunity was increased to 50%, **the ICER rose substantially.**"

3. *Line 180. Cohort characteristics. Did you assume a further cohort structure within these broad age bands? What did you assume about contact patterns between age or risk groups?*

Response: Our model categorized individuals into one of three age bands: <20 years, 20-59 years, and ≥60 years. We did not make any further assumptions about individuals within these age bands. These age groups generally align with strata of reported COVID-19 infection fatality ratios as well as with strategies adopted by South Africa's Department of Health, and other countries, to prioritize vaccination rollout. Our model assumes homogeneous mixing throughout the entire population regardless of age or risk group. We have added text to the Supplementary Methods to clarify this point.

Supplementary Methods, pages 24-25: **"Our model assumes that the force of infection is the same for all age groups (i.e., homogeneous mixing). This assumption is supported by two large seroprevalence studies in South Africa, which both demonstrated similar seroprevalence within age bands, both with community-based testing and with testing of blood donors [Sykes et al., Preprint; Hsiao et al., National Institute for Communicable Diseases, South Africa]."**

However, heterogeneous mixing can lead to protective effects in epidemic modeling, particularly in the case of vaccination and herd immunity. Thus our assumptions are most likely to result in conservative estimates of cost-effectiveness. We have added this to the Discussion.

Discussion, page 14: **"Our model assumes homogeneous mixing of the entire population. This assumption may result in conservative estimates of cost-effectiveness of vaccination, particularly at lower supply levels, because herd immunity is likely to be achieved at lower rates of vaccination after accounting for heterogeneous mixing [Britton et al., *Science* 2020]."**

4. *Pace of vaccination. It makes sense that the pace of vaccination is important (more people vaccinated and therefore protected within a particular time period) but I think that it really needs to be emphasised that what this really shows is the number of people that can be vaccinated relative to the next COVID-19 epidemic wave – i.e. what is assumed about pace is highly dependent on assumptions about future transmission.*

Response: We completely agree with the reviewer. To better highlight this issue we have added a new **Supplementary Figure 9**, which shows a two-way sensitivity analysis for R_e and pace of vaccination.

Supplementary Figure 9. Two-way sensitivity analysis of effective reproduction number (R_e) and vaccination pace: number of COVID-19 deaths averted by vaccination program.

We have also added an analysis which shows the influence of varying whether the next epidemic wave comes immediately, within one month, or within three months (results are in **Figure 1** and **Supplementary Table 3**). Vaccination was cost-saving in all three scenarios.

Additionally, given the increased attention to the delta variant of SARS-CoV-2, we have addressed how our analysis captures the potential influence of this variant.

Discussion, pages 11-12: “Much has been made about differences in the leading vaccine candidates and the impact of variants, such as the B.1.351 (**beta**) variant which eventually accounted for over 90% of SARS-CoV-2 infections in South Africa **and the B.1.617.2 (delta) variant**, on vaccine effectiveness. However, we found that, **even with substantially lower vaccine efficacy than reported in clinical trials, vaccination programs would prevent the majority of COVID-19 deaths compared to scenarios without vaccines.** For example, decreasing vaccine effectiveness against mild/moderate disease and severe/critical disease requiring hospitalization to 40% still reduced COVID-19 deaths by 65,800 (74%) compared with a scenario without vaccines. Although efficacy against symptomatic and severe disease have been the focus of vaccine trials, these parameters were less influential on population-wide health and cost outcomes than efficacy against infection, which is less commonly reported in trials. Nonetheless, the effectiveness ranges we examined in sensitivity analysis include the point estimates of efficacy against symptomatic and severe disease reported in clinical trials of the AstraZeneca ChAdOx1, Moderna mRNA-1273, and Pfizer-BioNTech mRNA BNT162b2 vaccines. This suggests that all of these vaccines are likely to have both health and economic benefits. **Furthermore, our sensitivity analysis examining different R_e scenarios likely captures the potential influence of more contagious SARS-CoV-2 variants such as delta.”**

5. *Supplementary Material, page 9 (vaccine efficacy calculations).* I did not interpret different endpoints of vaccine efficacy as being additive in this way. E.g. If vaccine efficacy against infection is 40%, and efficacy against disease is 60%, then the efficacy against disease in individuals who still become infected (your measure of $P(B)$ as I understood it) would be calculated as $1 - (1 - \text{efficacy_disease}) / (1 -$

efficacy_infection). Could you please clarify your reasoning for these equations?

Response: We appreciate the reviewer's attention to this. For clarity, we have redefined our equations to reflect the proportions among vaccinated individuals exposed to SARS-CoV-2 who are immune from infection, experience asymptomatic infection, or develop mild/moderate, severe, or critical disease (**Supplementary Methods pages 27-28**). We have also added a probability tree diagram to the Supplement (**Supplementary Figure 2**) to clarify our calculation of vaccine effectiveness and endpoint definitions.

Supplementary Methods, page 27: "We incorporated three measures of vaccine effectiveness into our model: effectiveness in preventing SARS-CoV-2 infection (VE_I), effectiveness in preventing **mild/moderate** COVID-19 disease (VE_M), and effectiveness in preventing severe or critical COVID-19 disease that would prompt hospitalization (VE_H)."

Supplementary Methods, pages 27-28: "**We accounted for the direct benefits of vaccination by adjusting disease path probabilities among those vaccinated. Disease path probabilities are age-dependent and defined as follows:**

- P_0 : probability immune to SARS-CoV-2 infection
- P_1 : probability of developing asymptomatic SARS-CoV-2 infection if exposed
- P_2 : probability of developing mild/moderate COVID-19 disease if exposed
- P_3 : probability of developing severe COVID-19 disease if exposed
- P_4 : probability of developing critical COVID-19 disease if exposed

Disease path probabilities for unvaccinated individuals susceptible to SARS-CoV-2 infection are shown in Supplementary Table 4. Among vaccinated individuals exposed to SARS-CoV-2, the probabilities of infection, mild/moderate disease, severe disease, or critical disease are functions of vaccine effectiveness and disease path probabilities among unvaccinated individuals exposed to SARS-CoV-2:

$$P(\text{immune}) = P_0^V = 1 - ((1 - VE_I)(1 - P_0)) \quad (4)$$

$$P(\text{asymptomatic infection}) = P_1^V = 1 - (P_0^V + P_2^V + P_3^V + P_4^V) \quad (5)$$

$$P(\text{mild/moderate disease}) = P_2^V = (1 - VE_M)P_2 \quad (6)$$

$$P(\text{severe disease}) = P_3^V = (1 - VE_H)P_3 \quad (7)$$

$$P(\text{critical disease}) = P_4^V = (1 - VE_H)P_4 \quad (8)$$

„

Supplementary Figure 2. Relationship between vaccine effectiveness and disease path probabilities among those vaccinated.

Age-dependent disease path probabilities

- P_0 : Probability immune from infection
- P_1 : Probability of developing asymptomatic infection if exposed
- P_2 : Probability of developing mild/moderate disease if exposed
- P_3 : Probability of developing severe disease if exposed
- P_4 : Probability of developing critical disease if exposed

Vaccine effectiveness measures

- VE_I : Vaccine effectiveness in preventing infection
- VE_M : Vaccine effectiveness in preventing mild/moderate disease
- VE_H : Vaccine effectiveness in preventing severe or critical disease requiring hospitalization

Minor comments.

6. *Abstract, line 68. Should this be “providing a vaccine” or “providing vaccination”.*

Response: We have modified this sentence.

Abstract, page 3: “Providing **vaccines** to at least 40% of the population and prioritizing vaccine rollout...”

7. *Line 196. Should this be “some individuals”?*

Response: We have modified this sentence.

Methods, page 19: “In the base case, we assumed that vaccine uptake among those eligible was 67%, accounting for vaccine hesitancy and failure to reach some **individuals**.”

8. *Results. First line – it would be helpful to restate here the analysis period over which you describe these results.*

Response: We have added the analysis period.

Results, page 6: “In both the $R_e=1.4$ scenario and the two-wave epidemic scenario, the absence of a vaccination program resulted in the most infections (~19-21 million) and deaths (70,400-89,300) and highest costs (~\$1.69-1.77 billion) **over the 360-day simulation period** (Table 1).”

9. *Line 332. “However, we found that even with lower effectiveness than that estimated from studies, a vaccination program would still prevent the majority of COVID-19 deaths that would occur without a vaccination program.” I am a bit unclear on what you mean in this sentence – when you say “majority of” if sounds like deaths are being prevented some other way? Perhaps consider rephrasing.*

Response: We have updated the sentence for clarity:

Discussion, pages 11-12: “However, we found that, even with **substantially lower vaccine efficacy than reported in clinical trials, vaccination programs would prevent the majority of COVID-19 deaths compared to scenarios without vaccines.**”

10. *Line 389, discussion section. Good to just briefly state here again what the uncertain parameters were?*

Response: We have now stated the uncertain parameters.

Discussion, pages 15: “**Primary assumptions in our model included initial prevalence of COVID-19, prevalence of prior immunity, time to vaccine rollout, and vaccine efficacy against asymptomatic infection.**”

11. Line 394. I think you mean to refer to “model simulations” here (or similar) – rather than “data”?

Response: We have modified the sentence as suggested.

Discussion, page 15: “Our **model simulations** underscore...”

Reviewer #2 (Remarks to the Author):

Major claim of the paper is that optimal COVID-19 vaccination program would reduce deaths and health care costs in South Africa. Implementation factors including procurement, distribution and pace of vaccine roll out are more influential than characteristics of the vaccine itself such as vaccine efficacy and price in maximizing public health benefits and economic efficiency.

The concept proposed and resulting interpretation is novel and of interest to others in the wider community.

Response: We thank the reviewer for their thorough consideration of our work and the wider impacts of our findings.

However, in terms of the conceptual framework and assumptions I have the following major comments.

12. First, the interpretation of the results given wide confidence intervals e.g. around R values etc should be presented with more care. Claiming 40% vaccination coverage is an optimal vaccination strategy in a South African setting can be harmful. The results have lots of uncertainties that heavily rely on assumptions with wide confidence intervals (vaccine efficacy while having variants of concern circulating, optimistic vaccine procurement prices etc) and based on a narrow scope of analyses.

Response: We thank the reviewer for raising this important concern and agree that outcomes under different levels of vaccination coverage are sensitive to assumptions regarding future dynamics of the epidemic. Our goal was not to determine an “optimal” level of coverage for South Africa, but to illustrate the relative impact that increasing vaccine supply or vaccination pace had on program cost-effectiveness. With this in mind, we have modified **Table 1** to include projected outcomes under the two-wave epidemic scenario. We have added results of the two-wave epidemic scenario to the Results.

Results, pages 6-7: “**In both the $R_e=1.4$ scenario and the two-wave epidemic scenario, the absence of a vaccination program** resulted in the most infections (~**19-21 million**) and deaths (**70,400-89,300**) and highest costs (~**\$1.69-1.77 billion**) over the 360-day simulation period (Table 1). Vaccinating 40% of the population decreased deaths (**82-85% reduction**) and resulted in the lowest total health care costs (**33-45% reduction**) in both scenarios. Increasing the vaccinated population to 67%, the government’s target for 2021, decreased deaths **and raised costs in both scenarios**. Increasing the vaccine supply to 80%, while simultaneously increasing vaccine acceptance to 80%, reduced deaths and raised costs **even** further in both scenarios. **In the $R_e=1.4$ scenario, the 67% supply strategy was less efficient (had a higher ICER) than the**

80% supply strategy, and the latter had an ICER of \$4,270/YLS compared with the 40% supply strategy. In the two-wave epidemic scenario, the 67% and 80% supply strategies had ICERs of \$1,990/YLS and \$2,600/YLS. A vaccine supply of 20%, while less efficient than higher vaccine supply levels, still reduced deaths by **72-76%** and reduced costs by **15-32%** compared with no vaccination. The highest vaccination pace, 300,000 vaccinations daily, resulted in the most favorable clinical outcomes and lowest costs compared with lower paces **in both the $R_e=1.4$ and the two-wave epidemic scenarios** (Table 1).

13. *There is no mentioning of equity concerns among different target groups in the analyses.*

Response: To highlight this issue, we have added to the Discussion.

Discussion, page 14: “Our modeled vaccination prioritization was based exclusively on age and not on employment type, comorbidity presence, or urban/rural heterogeneity in epidemiology or vaccination delivery. **Vaccination programs that target highly vulnerable groups are likely necessary to achieve most vaccination targets and have secondary benefits in terms of both population-level health outcomes and health equity goals.**”

14. *Also, vaccination strategies can be incrementally cost-effective but the decision makers in the current pandemic face broader questions, specifically on the health-non-health e.g. macroeconomic trade-offs. The incremental CE analyses does not take these issues into account. In the discussion the authors should highlight these limitations when interpreting the results for policy recommendations.*

Response: We have now highlighted this limitation in the Discussion.

Discussion, page 15: “**We assessed costs from a health care sector perspective and did not account for external costs associated with failure to achieve epidemic suppression (e.g., non-payment of wages and lost days of in-person education during periods of lockdown). Excluding these costs may underestimate the true value of COVID-19 vaccination to society.**”

15. *Second, related to the previous point. Disability is not included in the analyses which makes not only the \$/DALY threshold problematic but also not taking into account the impact of COVID-19 vaccines to younger/working populations relevant in a demographic setting like SA. The authors could refer to a recent paper that came out in Nature by Briggs and Vassall (see: <https://www.nature.com/articles/d41586-021-01392-2>)*

Response: We have now included this in the Discussion. Briggs and Vassall note that global DALYs attributable to COVID-19 are likely to be reported in the next burden of disease estimates from both the World Health Organization and the Institute for Health Metrics and Evaluation, and they also note that many low- and middle-income countries lack the reporting structure required to calculate QALYs and DALYs.

Discussion, page 14: “**Long-term disability among some of those who recover from COVID-19 is an important consideration for policymakers not captured by our model, which considers only years-of-life lost due to premature mortality. Our vaccination cost-effectiveness results may**

therefore be conservative, particularly among younger age groups that are less likely to die from COVID-19 but are still at risk for long-term sequelae [Briggs and Vassall, *Nature* 2021].”

16. *Third, also related to the long term effects of COVID why only looking at 1 year? What about vaccine effectiveness and duration protection? What about the sustainability of programs beyond one year? Would result look different beyond one year when public health and social measures (PHSM) are lifted?*

Response: We have added to the Discussion to address these important points.

Discussion, page 15: **“Given data limitations and the uncertainty in making long-term projections, we limited the time horizon of our analysis to one year. The sustainability and cost-effectiveness of vaccination beyond one year is likely to depend on the duration of protection conferred by existing vaccines, their effectiveness against emergent variants, and the costs, effectiveness, and frequency of potential booster shots—factors which remain unknown as of June 2021. If SARS-CoV-2 becomes endemic, cost-effectiveness analysis will become increasingly critical for integrating vaccination programs within health program budgets.”**

17. *Is it realistic in SA to reach coverage rates of over 60% in one year? What about the cost function to reaching difficult to reach populations?*

Response: We have modified the discussion of vaccination program costs to highlight these issues.

Discussion, page 12: “Similarly, we found that vaccination programs remained economically favorable even at relatively high vaccination costs. Though we did not explicitly account for all implementation and scale-up costs of a vaccination program, our estimates of cost per person vaccinated were based on reported costs of both vaccine and delivery in South Africa. **Achieving the government’s goal of vaccinating 67% of South Africans within one year will depend at least partially on global vaccine supplies and may require additional investments in vaccine procurement, delivery infrastructure, and community outreach. Although these expenses may increase program costs,** we found that the vaccination program would remain cost-saving at a vaccination cost up to \$25/person and likely cost-effective even at per-person vaccination cost up to \$75/person (ICER \$1,500/YLS). This is due to cost offsets in preventing hospitalizations.”

18. *Fourth, it is not clear what the authors assume in both the no vaccination and vaccination strategies with regards to other PHSM that are in place? Will these be lifted or remain the same when vaccination strategies are introduced? How will these influence the results.*

Response: We have now described this in the Discussion.

Discussion, page 15: **“We did not explicitly model the use of non-pharmaceutical interventions (NPIs) as a standalone strategy or in combination with vaccination. However, the evaluation of various transmission scenarios (including a sensitivity analysis in which R_0 changes over time) captures the potential impacts of different levels of NPI implementation on clinical outcomes.”**

Finally a couple of minor points.

19. *The authors claim that vaccine supply (and pace) can exert more direct control by policy makers (line 80 in supplement). Not sure this is true given the global vaccine supply shortage, specifically for LMICs. Also supply and access to vaccines also depends indirectly on procurement price of vaccines.*

Response: We agree that factors such as limited access to vaccines, supply interruptions, and vaccine procurement prices all present challenges that may cause LMICs to fall short of their vaccination goals. We have modified the sentence in the Supplement mentioned by the reviewer. We have also added to the Discussion to highlight the need for policymakers to take aggressive steps to ensure that supply and delivery goals are met.

Supplementary Methods, page 21: “In comparisons of different **levels of** vaccine supply and vaccination pace, the ICER of a given strategy was calculated...”

Discussion, page 12: “**Achieving the government’s goal of vaccinating 67% of South Africans within one year will depend at least partially on global vaccine supplies and may require additional investments in vaccine procurement, delivery infrastructure, and community outreach.**”

20. *Delivery costs from influenza focusing on elderly, PLWHA, ULMC etc are higher (5.42-8,22) to according to Edoka (2021) compared to the vaccine delivery costs per person (line 254 in supplement).*

Response: We appreciate the reviewer’s mention of the study of influenza vaccination cost-effectiveness in South Africa (Edoka et al., *Vaccine* 2021). We have reviewed this paper and found good agreement between the vaccine delivery cost used there and the cost used in our analysis, after adjusting for inflation and changes in the exchange rate between South African rand (ZAR) and United States dollars (USD). Specifically, after: (1) converting the vaccine delivery cost per dose used by Edoka et al. (\$5.42 [uncertainty range: \$2.71-\$8.13], expressed in 2018 USD) from 2018 USD to 2018 ZAR using average exchange rates for the period; (2) adjusting for inflation based on the change in South Africa’s consumer price index from 2018 to 2020; and (3) converting from 2020 ZAR to 2020 USD using exchange rates for this period, we obtain an influenza vaccine delivery cost of \$4.69 (uncertainty range: \$2.34-\$7.03), expressed in 2020 USD. This cost estimate is similar to the value used in our base case analysis (\$4.81, expressed in 2020 USD), which was derived from a retrospective analysis of KwaZulu-Natal’s HPV vaccination program (Moodley et al., *S Afr Med J* 2016).

REVIEWER COMMENTS

Reviewer #3 (Remarks to the Author):

Thank you for the opportunity to re-review the updates to this paper in lieu of Reviewer #2, with a focus on assessing whether Reviewer #2's comments have been taken into account by the authors.

Re reviewer comment 12, I appreciate the authors' clarification that 40% vaccination coverage is not an optimal vaccination strategy, as this seems to have been the reading of their preliminary results by South African policymakers, judging from my exposure to the South African decision making processes. I do believe that the clarifications in the current version are sufficient. Based on this I would like to suggest that it would be helpful if the authors could re-engage with South African policymakers (for example, the Ministerial Advisory Committee on COVID-19 Vaccines) in order to correct the initial impression.

Re comment 13, I do not agree with the authors' notion that "Vaccination programs that target highly vulnerable groups are likely necessary to achieve most vaccination targets". Even the lower vaccination targets, ie 40% uptake, modelled in this analysis could likely not be achieved by targeting the vulnerable groups mentioned- given the South African age distribution and the fact that, even though co-morbidities such as hypertension and diabetes mellitus Type II are highly prevalent, their diagnosis is not.

Re comment 14, the consideration of the "health-non-health e.g. macroeconomic trade-offs" that Reviewer #2 suggested is not the same as the factors the authors did add to the Discussion, ie, "non-payment of wages and lost days of in-person education during periods of lockdown". The macro-economic impact of the lockdowns in South Africa reaches far beyond unpaid wages and lost school days- these might be the concerns in a higher-income country. According to latest estimations, a net 1.2 million jobs have been lost¹ and South Africa's GDP has decreased by 7% in 2020/21², with direct and severe impacts on the overall government budget, including the health budget³. On the micro-economic level, a large percentage of South Africans have gone hungry during the last year. These factors, even though the authors might not have the methodological wherewithal to add them into the analysis, should at least be mentioned.

Re comment 19, I appreciate that the authors stress "the need for policymakers to take aggressive steps to ensure that supply and delivery goals are met", but it seems to me that reviewer #2 meant, or at least included, global policy makers in this, with the reference to the global vaccine supply shortage and the procurement price of vaccines, both of which are beyond the control of local policymakers in LMIC. It would strengthen the paper which otherwise focusses on describing and informing the South African (or LMIC) policy space, to add the perspective of global decision makers specifically.

Re comment 20, I follow the authors' justification of the cost of vaccine delivery. I would however suggest they similarly triangulate their cost of COVID-19 related hospitalisation to another paper by Edoka et al that has quantified the cost of ICU care for COVID-19 in South Africa based on ingredients costing. Edoka's analysis results in economic costs of USD 800-830 depending on the need for invasive vs. non-invasive ventilation. This is far lower than this paper's assumptions- a cost of USD 154 (77-309) per inpatient day for "hospitalization", apparently based on the tariff structure of a private hospital chain, and an (additional?) USD 1,751 (875-3,502) per day for ICU care, based on a cost analysis of ICU care (before COVID-19) in a public-sector hospital. Since the finding of cost savings across a number of the vaccination scenarios hinges on the cost of inpatient care, I would strongly suggest redoing the main analysis with a lower ICU cost informed by the Edoka paper, or at least include a value lower than 50% to the sensitivity analysis. It would also help if in the Methods section could be updated to a) clarify what "hospitalization costs" entail, if the ICU cost is additive to it, and to which populations they accrue, and b) if the original value is kept in the main analysis (which I really

don't think would make for a good base case), add a justification as to why a private-sector value is being used- given that throughout waves at least 50% of the admissions took place in the public sector.

References

1. StatsSA. Quarterly Labour Force Survey, Quarter 4: 2020. Pretoria: StatsSA, 2021. URL: <http://www.statssa.gov.za/publications/P0211/P02114thQuarter2020.pdf>
2. National Treasury. Budget Review, National Treasury, Pretoria: February 2021. URL: [http://www.treasury.gov.za/documents/national budget/2021/default.aspx](http://www.treasury.gov.za/documents/national%20budget/2021/default.aspx)
3. National Treasury. Provincial Estimates of Revenue and Expenditure, 2021. Pretoria: National Treasury, 2021.
4. Wills G, Patel L, Van der Berg S et al.. Household resource flows and food poverty during South Africa's lockdown: Short-term policy implications for three channels of social protection. NIDS CRAM Wave 1. 2020 www.cramsurvey.org

RESPONSES TO REVIEWER COMMENTS

Page numbers refer to the tracked versions of the revised manuscript and supplement.

Reviewer #3:

2. Re reviewer comment 12, I appreciate the authors' clarification that 40% vaccination coverage is not an optimal vaccination strategy, as this seems to have been the reading of their preliminary results by South African policymakers, judging from my exposure to the South African decision making processes. I do believe that the clarifications in the current version are sufficient. Based on this I would like to suggest that it would be helpful if the authors could re-engage with South African policymakers (for example, the Ministerial Advisory Committee on COVID-19 Vaccines) in order to correct the initial impression.

Response: We thank the Reviewer for this suggestion to re-engage with South African policymakers regarding our findings. This is a high priority for us, and we are already planning to do so. Among the co-authors of this work is Dr. Richard Lessells, a member of the Ministerial Advisory Committee on COVID-19 Vaccines in South Africa. Dr. Lessells will engage with colleagues on the Committee to facilitate a presentation of our final report, once accepted for publication.

3. Re comment 13, I do not agree with the authors' notion that "Vaccination programs that target highly vulnerable groups are likely necessary to achieve most vaccination targets". Even the lower vaccination targets, ie 40% uptake, modelled in this analysis could likely not be achieved by targeting the vulnerable groups mentioned- given the South African age distribution and the fact that, even though co-morbidities such as hypertension and diabetes mellitus Type II are highly prevalent, their diagnosis is not.

Response: We recognize that our statement could be misleading. We have clarified this in the Discussion.

Discussion, page 13: "Vaccination programs that **reach vulnerable and disadvantaged** groups **would likely improve** population-level health outcomes and health equity."

4. Re comment 14, the consideration of the "health-non-health e.g. macroeconomic trade-offs" that Reviewer #2 suggested is not the same as the factors the authors did add to the Discussion, ie, "non-payment of wages and lost days of in-person education during periods of lockdown". The macro-economic impact of the lockdowns in South Africa reaches far beyond unpaid wages and lost school days- these might be the concerns in a higher-income country. According to latest estimations, a net 1.2

million jobs have been lost¹ and South Africa's GDP has decreased by 7% in 2020/21², with direct and severe impacts on the overall government budget, including the health budget³. On the micro-economic level, a large percentage of South Africans have gone hungry during the last year. These factors, even though the authors might not have the methodological wherewithal to add them into the analysis, should at least be mentioned.

Response: The Reviewer raises very important points about the broader impact of the pandemic and the associated lockdown. As noted by the Reviewer, it is difficult to capture all these important factors within a model. Even without accounting for these factors, our results provide support, on clinical and health economic grounds, for broadening and expediting vaccination programs. Including these factors would bolster the argument for vaccination programs. We have modified the Discussion.

Discussion, page 13: "We assessed costs from a health care sector perspective and did not account for **other sector** costs associated with **lockdowns and** failure to achieve epidemic suppression (e.g., **macroeconomic factors such as job and productivity losses and microeconomic factors such as reduced household income and disruptions to education**)."

We thank the reviewer for the references, two of which we have added to this section.

5. Re comment 19, I appreciate that the authors stress "the need for policymakers to take aggressive steps to ensure that supply and delivery goals are met", but it seems to me that reviewer #2 meant, or at least included, global policy makers in this, with the reference to the global vaccine supply shortage and the procurement price of vaccines, both of which are beyond the control of local policymakers in LMIC. It would strengthen the paper which otherwise focusses on describing and informing the South African (or LMIC) policy space, to add the perspective of global decision makers specifically.

Response: We agree that action from global policymakers is essential to combatting the global vaccine supply shortage. We have now highlighted this.

Discussion, page 11: "Achieving the government's goal of vaccinating 67% of South Africans within one year will depend at least partially on global vaccine supplies and may require **global policymakers to better fund and facilitate vaccine distribution and accessible pricing for LMICs, in addition to local attention to** delivery infrastructure and community outreach."

We have written another paper, currently under review, that examines the cost-effectiveness of COVID-19 vaccination across 91 low- and middle-income countries from a global funding perspective (preprint available on medRxiv, <https://doi.org/10.1101/2021.04.28.21256237>).

6. Re comment 20, I follow the authors' justification of the cost of vaccine delivery. I would however suggest they similarly triangulate their cost of COVID-19 related hospitalisation to another paper by Edoaka et al that has quantified the cost of ICU care for COVID-19 in South Africa based on ingredients costing. Edoaka's analysis results in economic costs of USD 800-830 depending on the need for invasive vs. non-invasive ventilation. This is far lower than this paper's assumptions- a cost of USD 154 (77-309) per inpatient day for "hospitalization", apparently based on the tariff structure of a private hospital chain, and an (additional?) USD 1,751 (875-3,502) per day for ICU care, based on a cost analysis of ICU care (before COVID-19) in a public-sector hospital. Since the finding of cost savings across a number of the

vaccination scenarios hinges on the cost of inpatient care, I would strongly suggest redoing the main analysis with a lower ICU cost informed by the Edoka paper, or at least include a value lower than 50% to the sensitivity analysis. It would also help if in the Methods section could be updated to a) clarify what “hospitalization costs” entail, if the ICU cost is additive to it, and to which populations they accrue, and b) if the original value is kept in the main analysis (which I really don’t think would make for a good base case), add a justification as to why a private-sector value is being used- given that throughout waves at least 50% of the admissions took place in the public sector.

Response: We appreciate the Reviewer bringing to our attention the paper by Edoka et al. We have now completed additional analyses in which the costs of non-ICU hospital care and ICU care reflect those reported by Edoka et al. - \$137/day for non-ICU general ward care and \$798/day for ICU care, not including facility fees. Notably, this ICU care cost estimate is 46% of our base case value, which is nearly identical to our prior sensitivity analysis that included hospital and ICU care costs of 50% of the base case values. We have mentioned this new analysis in the Results and Supplementary Methods and added a Supplementary Table 4, similar to Table 1, with these results. Though some ICERs increased, the general conclusions were similar to those of our prior analysis: vaccine supplies of 40% or 80% were non-dominated (with the latter providing greater clinical benefit), while a faster vaccination pace resulted in greater clinical benefit and lower costs.

We would also like to note that while the non-ICU general ward care cost used in our base case analysis (\$154 per day) is reported by Netcare, a private hospital network, this cost is similar to the non-ICU general ward care cost in the public sector reported by Edoka et al. (\$137 per day excluding facility fee, \$202 per day including facility fee). The high care ward daily cost reported by Edoka et al. was \$278 excluding facility fee and \$474 including facility fee, both higher than the cost in our base case analysis. If we were to apply the high care ward daily cost from Edoka et al. as the daily non-ICU hospital care cost in our model analysis, then the vaccination program would have a lower ICER (i.e., would be more cost-effective) than we have shown.

The ICU care cost used in our base case analysis (\$1,751 per day) is from a cost analysis of a public hospital in South Africa by Mahomed et al. The Reviewer notes the public sector ICU cost reported by Edoka et al. (“USD 800-830”). However, this cost excludes the facility fee. When Edoka et al. included the facility fee, the ICU care cost was \$1,314 per day, closer to our base case value and well within the range of our previously conducted sensitivity analysis.

Although the precise costs of hospital and ICU care are variable and depend on services needed, our intent is to provide a range of estimates that enable readers to draw inferences about the cost-effectiveness of the vaccination program in South Africa. We feel that this updated analysis with lower ICU cost assumptions, along with our sensitivity analyses, permit that.

Results, page 8: “When several of the main analyses were repeated with lower costs of hospital and ICU care, some ICERs increased, but vaccine supplies of 40% or 80% remained non-dominated (with the latter providing greater clinical benefit), while a faster vaccination pace still resulted in greater clinical benefit and lower costs (Supplementary Table 4).”

Methods, page 19: “We applied vaccination costs and daily costs of hospital care and ICU care based on published estimates and/or cost quotes obtained in South Africa (Table 3 and Supplementary Methods).”

Table 3, footnote: **“^bThe range of ICU care costs includes the cost (from Edoaka et al.) applied in a repeat of several of the main analyses.”**

Supplement, pages 32-33: **“In the base case, we applied a daily cost of \$154 for hospital (non-ICU) ward care based on cost per inpatient day in a general ward charged by Netcare, a private hospital network. Those in the “severe” and “recuperation” states accrued this daily cost, assuming a hospital ward bed was available. In the base case, we applied a daily cost of \$1,751 for ICU care based on a cost analysis of a public hospital in KwaZulu-Natal, South Africa. Those in the “critical” state accrued this daily cost and not the hospital ward care cost, assuming an ICU bed was available. We performed sensitivity analysis in which we varied the hospital and ICU care costs to 50% and 200% of the base case values.**

Additionally, we repeated the main analysis using costs reported by Edoaka et al. in the public sector. These were \$137/day for hospital (non-ICU) general ward care with supplemental oxygen and \$798/day for ICU care with invasive mechanical ventilation, both excluding facility fees (when including facility fees, Edoaka et al. reported costs of \$202/day for hospital ward care and \$1,314/day for ICU care). The results of this analysis are in Supplementary Table 4.”

New Supplementary Table 4: please see the next page.

Supplementary Table 4. Clinical and economic outcomes of different COVID-19 vaccination program strategies of vaccine supply and vaccination pace under different scenarios of epidemic growth in South Africa: applying lower costs of hospital and ICU care from Edoaka et al.

Scenario and Vaccination Strategy	Cumulative SARS-CoV-2 infections	Cumulative COVID-19 deaths	Years-of-life lost	Health care costs, USD	ICER, USD per year-of-life saved ^a
Vaccine supply					
$R_e = 1.4$					
Vaccine supply 40%	11,784,700	16,000	275,800	797,871,600	--
Vaccine supply 20%	15,489,500	21,800	397,300	890,934,300	Dominated
Vaccine supply 67%	10,585,100	14,700	259,600	993,638,400	Dominated
No vaccination	21,012,100	89,300	1,558,700	1,005,281,800	Dominated
Vaccine supply 80% ^b	10,410,000	12,000	217,900	1,092,247,100	5,080
Two-wave epidemic^c					
Vaccine supply 40%	7,758,800	10,600	175,100	658,380,800	--
Vaccine supply 20%	12,765,900	19,900	371,500	702,362,400	Dominated
Vaccine supply 67%	5,594,000	7,800	133,700	810,775,600	3,680
Vaccine supply 80% ^b	5,940,500	6,900	119,100	891,054,000	5,480
No vaccination	19,290,400	70,400	1,206,200	948,370,400	Dominated
Vaccination pace					
$R_e = 1.4$					
Pace 300,000 vaccinations per day	5,659,400	7,200	120,300	814,977,100	--
Pace 200,000 vaccinations per day	8,191,900	9,600	151,300	876,892,500	Dominated
Pace 150,000 vaccinations per day	10,585,100	14,700	259,600	993,638,400	Dominated
No vaccination	21,012,100	89,300	1,558,700	1,005,281,800	Dominated
Two-wave epidemic^c					
Pace 300,000 vaccinations per day	2,697,100	3,200	49,300	688,458,100	--
Pace 200,000 vaccinations per day	4,148,500	5,900	90,300	744,446,000	Dominated
Pace 150,000 vaccinations per day	5,594,000	7,800	133,700	810,775,600	Dominated
No vaccination	19,290,400	70,400	1,206,200	948,370,400	Dominated

Supplementary Table 4, continued.

Scenario and Vaccination Strategy	Cumulative SARS-CoV-2 infections	Cumulative COVID-19 deaths	Years-of-life lost	Health care costs, USD	ICER, USD per year-of-life saved ^a
Vaccine supply and vaccination pace					
$R_e = 1.4$					
Vaccine supply 40%, pace 300,000 vaccinations per day	9,866,800	13,000	211,300	688,324,400	--
Vaccine supply 40%, pace 150,000 vaccinations per day	11,784,700	16,000	275,800	797,871,600	Dominated
Vaccine supply 67%, pace 300,000 vaccinations per day	5,659,400	7,200	120,300	814,977,100	1,390
Vaccine supply 67%, pace 150,000 vaccinations per day	10,585,100	14,700	259,600	993,638,400	Dominated
No vaccination	21,012,100	89,300	1,558,700	1,005,281,800	Dominated
Two-wave epidemic^c					
Vaccine supply 40%, pace 300,000 vaccinations per day	6,223,600	7,200	126,900	580,478,200	--
Vaccine supply 40%, pace 150,000 vaccinations per day	7,758,800	10,600	175,100	658,380,800	Dominated
Vaccine supply 67%, pace 300,000 vaccinations per day	2,697,100	3,200	49,300	688,458,100	1,390
Vaccine supply 67%, pace 150,000 vaccinations per day	5,594,000	7,800	133,700	810,775,600	Dominated
No vaccination	19,290,400	70,400	1,206,200	948,370,400	Dominated

ICU: intensive care unit. USD: United States dollars. ICER: incremental cost-effectiveness ratio. R_e : effective reproduction number. Dominated: the strategy results in a higher ICER than that of a more clinically effective strategy, or the strategy results in less clinical benefit (more years-of-life lost) and higher health care costs than an alternative strategy.

In this analysis, daily costs of hospital care (\$137) and ICU care (\$798) were applied from a report by Edoaka et al.¹ These values did not include facility costs.

^aWithin each R_e scenario, vaccination strategies are ordered from lowest to highest cost per convention of cost-effectiveness analysis. ICERs are calculated compared to the next least expensive, non-dominated strategy. Displayed life-years and costs are rounded to the nearest hundred, while ICERs are calculated based on non-rounded life-years and costs and then rounded to the nearest ten.

^bWhen modeling a vaccination program that seeks to vaccinate 80% of the population, uptake among those eligible was increased to 80% to avoid a scenario in which supply exceeds uptake. If uptake is not increased beyond 67%, then the strategy of vaccinating 67% of the population provides the most clinical benefit and results in an ICER of \$12,110/YLS compared with vaccinating 40% of the population when R_e is 1.4 and \$3,680/YLS in an epidemic scenario with periodic surges.

^cIn the analysis of an epidemic with periodic surges, the basic reproduction number (R_0) alternates between low and high values over time, and the R_e changes day-to-day as the epidemic and vaccination program progress and there are fewer susceptible individuals. For most of the simulation horizon, R_0 is 1.6 (equivalent to an initial R_e of 1.1). However, during days 90-150 and 240-300 of the simulation, R_0 is increased to 2.6. This results in two epidemic waves with peak R_e of approximately 1.4-1.5.

REVIEWERS' COMMENTS

Reviewer #3 (Remarks to the Author):

Thank you to the authors for your thorough consideration of my comments. They have all been sufficiently taken into account.